# Robust Watermarking Using Generative Priors Against Image Editing: From Benchmarking to Advances

**Shilin Lu[1], Zihan Zhou[1], Jiayou Lu[1], Yuanzhi Zhu[2], Adams Wai-Kin Kong[1]**
[1]Nanyang Technological University, [2]ETH Zürich
{shilin002, zihan010, jiayou001}@e.ntu.edu.sg
zyzeroer@gmail.com, adamskong@ntu.edu.sg

## Abstract

Current image watermarking methods are vulnerable to advanced image editing techniques enabled by large-scale text-to-image models. These models can distort embedded watermarks during editing, posing significant challenges to copyright protection. In this work, we introduce **W-Bench**, the first comprehensive benchmark designed to evaluate the robustness of watermarking methods against a wide range of image editing techniques, including image regeneration, global editing, local editing, and image-to-video generation. Through extensive evaluations of eleven representative watermarking methods against prevalent editing techniques, we demonstrate that most methods fail to detect watermarks after such edits. To address this limitation, we propose **VINE**, a watermarking method that significantly enhances robustness against various image editing techniques while maintaining high image quality. Our approach involves two key innovations: (1) we analyze the frequency characteristics of image editing and identify that blurring distortions exhibit similar frequency properties, which allows us to use them as surrogate attacks during training to bolster watermark robustness; (2) we leverage a large-scale pretrained diffusion model SDXL-Turbo, adapting it for the watermarking task to achieve more imperceptible and robust watermark embedding. Experimental results show that our method achieves outstanding watermarking performance under various image editing techniques, outperforming existing methods in both image quality and robustness. Code is available at https://github.com/Shilin-LU/VINE.

## 1 Introduction

The primary function of an image watermark is to assert copyright or verify authenticity. A key aspect of watermark design is ensuring its robustness against various image manipulations. Prior deep learning-based watermarking methods (Bui et al., 2023; Tancik et al., 2020; Zhu, 2018) have proven effective at withstanding classical transformations (e.g., compression, noising, scaling, and cropping). However, recent advances in large scale text-to-image (T2I) models (Chang et al., 2023; Ramesh et al., 2022; Rombach et al., 2022; Saharia et al., 2022) have significantly enhanced image editing capabilities, offering a wide array of user-friendly manipulation tools (Brooks et al., 2023; Zhang et al., 2024b). These T2I-based editing methods produce highly realistic modifications, rendering the watermark nearly undetectable in the edited versions. This poses challenges for copyright and intellectual property protection, as malicious users can easily alter an artist's or photographer's work, even with embedded watermarks, to create new content without proper attribution.

In this work, we present **W-Bench**, the first holistic benchmark that incorporates four types of image editing techniques to assess the robustness of watermarking methods, as shown in Figure 1(a). Eleven representative watermarking methods are evaluated on W-Bench. The benchmark encompasses image regeneration, global editing, local editing, and image-to-video generation (I2V). (1) **Image regeneration** involves perturbing an image into a noisy version and then reconstructing it, which can be categorized as either stochastic (Meng et al., 2021; Zhao et al., 2023b) or deterministic (also known as image inversion) (Mokady et al., 2022; Song et al., 2020a). (2) For **global**

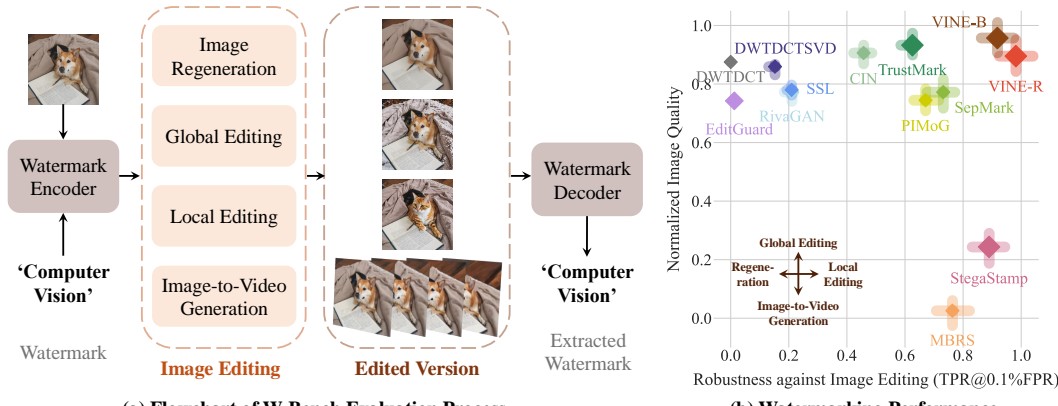

**(a) Flowchart of W-Bench Evaluation Process**    **(b) Watermarking Performance**

Figure 1: (a) Flowchart of the W-Bench evaluation process. (b) Watermarking performance. Each method is illustrated with a diamond and four bars. The area of the diamond represents the method's encoding capacity. The y-coordinate of the diamond's center indicates normalized image quality, calculated by averaging the normalized PSNR, SSIM, LPIPS, and FID between watermarked and input images. The x-coordinate represents robustness, measured by the True Positive Rate at a 0.1% False Positive Rate (TPR@0.1%FPR) averaged across four types of image editing methods, encompassing a total of seven distinct models and algorithms. The four bars are oriented to signify different editing tasks: image regeneration (left), global editing (top), local editing (right), and image-to-video generation (bottom). The length of each bar reflects the method's normalized TPR@0.1%FPR after each type of image editing—the longer the bar, the better the performance.

**editing**, we use models such as Instruct-Pix2Pix (Brooks et al., 2023) and MagicBrush (Zhang et al., 2024b), which take the image and a text prompt as inputs to edit images. (3) For **local editing**, we employ models like ControlNet-Inpainting (Zhang et al., 2023) and UltraEdit (Zhao et al., 2024c), which allow an additional mask input to specify the region to be modified. (4) Additionally, we evaluate watermarking models in the context of **image-to-video generation** using Stable Video Diffusion (SVD) (Blattmann et al., 2023) to determine whether the watermark remains detectable in the resultant video frames. Although this is not a conventional image editing approach, we consider it a special case that allows us to identify if the generated videos use copyrighted images. Experimental results (Figure 1(b)) reveal that most previous watermarking models struggle to extract watermarks after images are edited by these methods. StegaStamp (Tancik et al., 2020) and MBRS (Jia et al., 2021) manage to retain watermarks in certain cases, but at the expense of image quality.

To this end, we propose **VINE**, an invisible watermarking model designed to be robust against image editing. Our improvements focus on two key components: the **noise layers** and the **watermark encoder**. For the noise layers, a straightforward way to train a watermarking model robust to image editing would be to incorporate editing processes into the training pipeline. However, this is nearly infeasible for large scale T2I model-based image editing because it requires backpropagating through the entire sampling process, which can lead to memory issues (Salman et al., 2023). Instead, we seek surrogate attacks by analyzing image editing from a frequency perspective. The key insight is that image editing tends to remove patterns embedded in high-frequency bands, while those in low-frequency bands are less affected. This property is also observed in blurring distortions (e.g., pixelation and defocus blur). The experiments show that incorporating various blurring distortions into the noise layers can enhance the robustness of the watermarking against image editing.

However, this robustness comes at the cost of watermarked image quality, which is limited by the capability of the watermark encoder. To address this, we leverage a large-scale pretrained generative model, such as SDXL-Turbo (Sauer et al., 2023), as a powerful generative prior, adapting it specifically for the watermarking task. In this context, the watermark encoder functions as a conditional generative model, taking original images and watermarks as inputs and generating watermarked images with a distinct distribution that can be reliably recognized by the corresponding decoder. By utilizing this strong generative prior, the watermark is embedded more effectively, resulting in both improved perceptual image quality and enhanced robustness.

Our contributions are summarized as follows:

1. We present W-Bench, the first comprehensive benchmark designed to evaluate eleven representative watermarking models across various image editing methods: image regeneration,

      global editing, local editing, and image-to-video generation. This evaluation covers seven widely used editing models and algorithms and demonstrates that current watermarking models are vulnerable to them.

2. We reveal that image editing predominantly removes watermarking patterns in high-frequency bands, while those in low-frequency bands remain less affected. This phenomenon is also observed in certain types of blurring distortion. Such distortions can be used as surrogate attacks to overcome the challenges from using T2I models during training and to enhance the robustness of the watermark.

3. We approach the watermark encoder as a conditional generative model and introduce two techniques to adapt SDXL-Turbo, a pretrained one-step text-to-image model, for the watermarking task. This powerful generative prior improves both the perceptual quality of watermarked images and their robustness to various image editing. Experimental results demonstrate that our model, VINE, is robust against multiple image editing methods while maintaining high image quality, outperforming existing watermarking models.

## 2 RELATED WORK

**Watermarking benchmark.** To the best of our knowledge, WAVES (An et al., 2024) is currently the only comprehensive benchmark for evaluating the robustness of deep learning-based watermarking methods against image manipulations driven by large-scale generative models. However, WAVES considers only image regeneration (Zhao et al., 2023b) among prevalent image editing techniques and does not include other T2I-based editing models. In contrast, W-Bench encompasses not only image regeneration but also global editing (Brooks et al., 2023), local editing (Zhang et al., 2023), and image-to-video generation (Blattmann et al., 2023), thereby broadening the scope of our assessment of image editing methods. Furthermore, WAVES evaluates only three watermarking methods—StegaStamp (Tancik et al., 2020), Stable Signature (Fernandez et al., 2023), and Tree-Ring (Wen et al., 2023). Notably, Stable Signature and Tree-Ring are limited to generated images and cannot be applied to real ones. In contrast, W-Bench is designed to evaluate watermarking models that work with any type of image, thereby enhancing their effectiveness for copyright protection.

**Robust watermarking.** Watermarking has long been studied for purposes such as tracking and protecting intellectual property in images (Al-Haj, 2007; Cox et al., 2007; Navas et al., 2008), videos (Fernandez et al., 2024; Li et al., 2024), and 3D assets (Ren et al., 2025). Recently, deep learning-based methods (Bui et al., 2023; Chen & Li, 2024; Fang et al., 2022; 2023; Jia et al., 2021; Kishore et al., 2021; Luo et al., 2020; 2024; Ma et al., 2022; Tancik et al., 2020; Wu et al., 2023; Zhu, 2018; Zhang et al., 2019; 2021), have demonstrated competitive robustness against a wide range of transformations. However, these methods remain vulnerable to image editing powered by large-scale generative models. Three recent studies—EditGuard (Zhang et al., 2024d), Robust-Wide (Hu et al., 2024), and JigMark (Pan et al., 2024)—have begun to develop watermarking models that are robust to such image editing. However, although EditGuard, which replaces the unedited regions of an edited image with corresponding areas of the unedited watermarked version, demonstrates good performance, its application in real-world scenarios could be ineffective if the detector has no access to the unedited watermarked image. In our benchmark, we assume that the detector has access only to the edited image without any additional information. JigMark employs contrastive learning to train a watermark encoder and a classifier that determines if an image is watermarked, but it does not support decoding free-form messages. Robust-Wide incorporates Instruct-Pix2Pix into the noise layer using gradient truncation, but its generalization to other editing models could be less effective.

## 3 METHOD

Given an original image $x_o$ and a watermark $w$, our goal is to imperceptibly embed the watermark into the image by an encoder $E(\cdot)$ to obtain a watermarked image $x_w = E(x_o, w)$. The watermark should be accurately extracted by a corresponding decoder $D(\cdot)$ from $x_w$, i.e., $w' = D(x_w)$, even when $x_w$ undergoes image editing $\epsilon(\cdot)$.

In Section 3.1, we examine the frequency properties of various image editing methods and identify surrogate attacks that enhance the robustness of watermarking against them. In Section 3.2, we further improve both the robustness and quality of the watermarked image by adapting a one-step

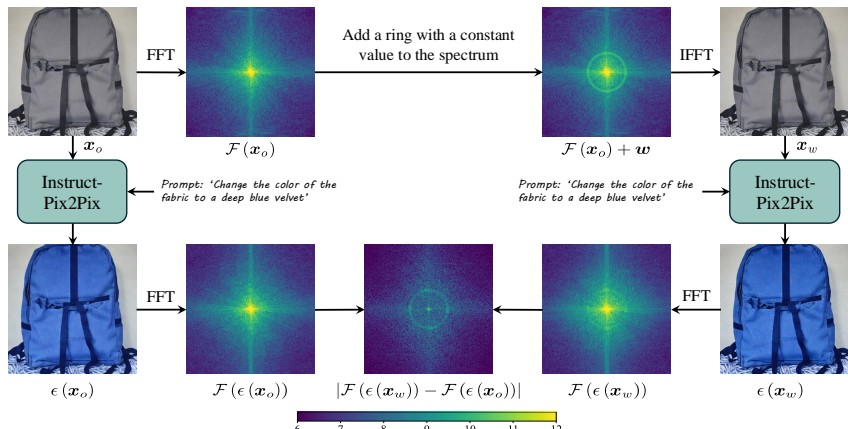

Figure 2: Process for analyzing the impact of image editing on an image's frequency spectrum. In this example, the editing model Instruct-Pix2Pix, denoted as $\epsilon(\cdot)$, is employed. The function $\mathcal{F}(\cdot)$ represents the Fourier transform, and we visualize its magnitude on a logarithmic scale.

text-to-image model as the watermark encoder. Additionally, we introduce several techniques to facilitate this adaptation. Section 3.3 details the training loss functions and strategies employed, as well as a resolution scaling method used in our experiments.

### 3.1 FREQUENCY NATURE OF IMAGE EDITING

To develop a robust watermarking model against image editing, a straightforward way is to integrate image editing models in the noise layers between the encoder and decoder during training. However, many prevalent image editing methods are based on diffusion models, which typically involve multiple sampling steps to produce edited images. This can lead to memory issues when backpropagating through the denoising process. Alternative methods, such as gradient truncation (Hu et al., 2024; Yuan et al., 2024), achieve subpar results, and the straight-through estimator (Bengio et al., 2013) fails to converge when training from scratch. Thus, we seek surrogate attacks during training.

We start by examining how image editing methods influence the spectrum of an image. Specifically, we conduct three sets of experiments in which symmetric patterns are inserted into the low-, mid-, and high-frequency bands. Figure 2 illustrates the analysis process for a pattern inserted into the low-frequency band. In this procedure, a ring-shaped pattern, $w$, with a constant value is embedded in the low-frequency region of the Fourier spectra of the RGB channels of the original image, $x_o$, i.e., $\mathcal{F}(x_o) + w$. The inverse Fourier transform is then applied to obtain the watermarked one, $x_w$. The image editing model, denoted as $\epsilon(\cdot)$, is applied to both the original image $x_o$ and the watermarked one $x_w$, producing edited versions $\epsilon(x_o)$ and $\epsilon(x_w)$, respectively. Finally, we compute the difference between their Fourier spectra, $|\mathcal{F}(\epsilon(x_w)) - \mathcal{F}(\epsilon(x_o))|$, to assess how the inserted pattern is affected by the editing process. The details of the used image editing methods are provided in Section 4.1.

Figure 3 illustrates that image editing methods typically remove patterns in the mid and high-frequency bands, while low-frequency patterns remain relatively unaffected. This suggests that T2I-based image editing methods often fail to reproduce intricate mid and high-frequency details. We infer that this occurs because T2I models are trained to prioritize capturing the overall semantic content and structure of images (i.e., primarily the low-frequency components) for aligning with the text prompts. As a result, high-frequency patterns are smoothed out during the generation process.

To develop a robust watermarking model against image editing, it should learn to embed information into the low-frequency bands. To identify effective surrogate attacks, we explore various image distortions, denoted as $\mathcal{T}(\cdot)$, that resemble image editing to some extent. Both image editing and distortion methods preserve the overall image layout and most of the content, although image distortions typically result in lower perceptual quality. Notably, as shown in Figure 3, certain blurring distortions (e.g., pixelation and defocus blur) exhibit a similar trend to image editing. In contrast, widely used distortions, such as JPEG compression and saturation, do not display this behavior. Since these blurring distortions are computationally efficient, we incorporate them with varying severity levels into our noise layers during training. This encourages the model to embed information within the low-frequency bands (see the frequency patterns of each watermarking method in Appendix B). Consequently, the robustness against image editing is enhanced, as demonstrated by

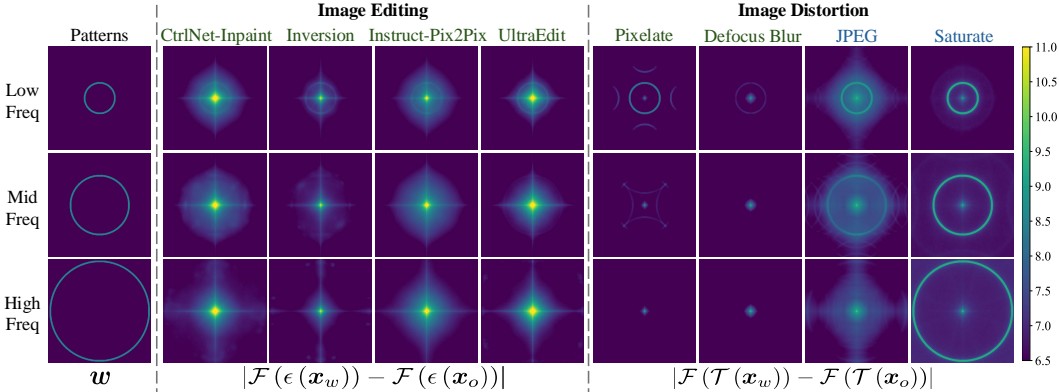

Figure 3: Impact of various image editing techniques and distortions on the frequency spectra of images. Results are averaged over 1,000 images. Image editing methods tend to remove frequency patterns in the mid- and high-frequency bands, while low-frequency patterns remain largely unaffected. This trend is also observed with blurring distortions such as pixelation and defocus blur. In contrast, commonly used distortions like JPEG compression and saturation do not exhibit similar behavior in the frequency domain. The analysis of SVD is not included, as it removes all patterns, rendering them invisible to the human eye. A discussion on SVD can be found in Section 4.3.

the ablation study in Table 2. The complete set of distortions applied in the noise layer includes common distortions such as saturation, contrast, brightness adjustments, JPEG compression, Gaussian noise, shot noise, impulse noise, and speckle noise to counteract image degradation caused by transmission. Additionally, we incorporate various blurring distortions—such as pixelation, defocus blur, zoom blur, Gaussian blur, and motion blur—to resist image editing.

## 3.2 GENERATIVE PRIOR FOR WATERMARK ENCODING

Although incorporating image distortions into the noise layers can enhance robustness against image editing, this improvement comes at the expense of watermarked image quality, which is constrained by the capabilities of the watermark encoder. The watermark encoder can be viewed as a conditional generative model, where the conditions include both a watermark and a detailed image, rather than simpler representations like depth maps, Canny edges, or scribbles. We hypothesize that a powerful generative prior can facilitate embedding information more invisibly while enhancing robustness. Thus, we aim to adapt a large-scale T2I model as a watermark encoder. There are two types of large scale T2I models: multi-step and one-step. Multi-step T2I models complicate the backpropagation of watermark extraction loss and suffer from slow inference speeds. Thus, we use a one-step pre-trained text-to-image model, SDXL-Turbo (Sauer et al., 2023).

To convert SDXL-Turbo into a watermark encoder, an effective strategy to incorporate both the input image and the watermark into the model is essential. A common strategy for integrating conditions into diffusion models is to introduce additional adapter branches (Mou et al., 2024; Zhang et al., 2023). However, in the one-step generative model, the noise map—the input to the UNet—directly determines the final layout of the generated images (Sauer et al., 2023). This contrasts with multi-step diffusion models, where the image layout is gradually established during the early sampling stages. Adding an extra conditional branch to the one-step model causes the UNet to receive two sets of residual features, each representing distinct structures. This makes the training process more challenging and results in subpar performance, as demonstrated by the ablation study in Table 2. Instead, as illustrated in Figure 4, we employ a condition adaptor to fuse the information from the input image and the watermark (the architecture of the condition adaptor is shown in Figure 11). This fused data is then fed into the VAE encoder to obtain latent features, which are subsequently input into the UNet and VAE decoder to generate the final watermarked image. We also attempted to input the watermark through the text prompt and finetune the text encoder simultaneously, but this approach failed to converge. Thus, during training, the text prompt is set to a null prompt.

Despite the general effectiveness of SDXL-Turbo's VAE, its architecture is not ideally suited for watermarking tasks. The VAE is designed to balance reconstructive and compressive capabilities, thus the reconstructive fidelity is traded off for smoother latent spaces and better compressibility. In the context of watermarking, however, the reconstruction capability is crucial to make sure that

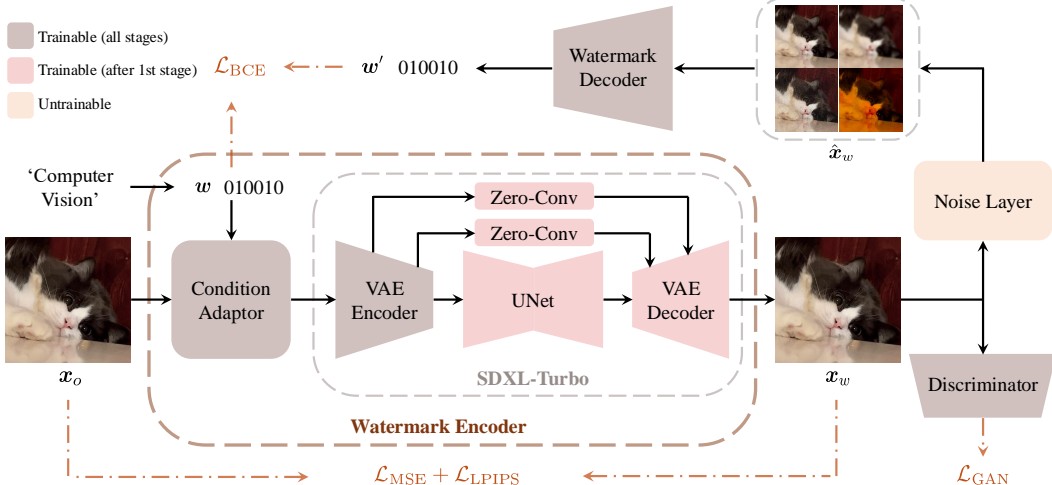

Figure 4: The overall framework of our method, **VINE**. We utilize the pretrained one-step text-to-image model SDXL-Turbo as the watermark encoder. A condition adaptor is incorporated to fuse the watermark with the image before passing the information to the VAE encoder. Zero-convolution layers (Zhang et al., 2023) and skip connections are added for better perceptual similarity. For decoding the watermark, we employ ConvNeXt-B (Liu et al., 2022b) as the decoder, with an additional fully connected layer to output a 100-bit watermark. Throughout the entire training process, the SDXL-Turbo text prompt is set to null prompt. Figure 11 shows the condition adaptor architecture.

the watermarked image is perceptually identical to the input image. Thus, we enhance the VAE by introducing skip connections between the encoder and decoder (Figure 4). Concretely, we extract four intermediate activations after each downsampling block in the encoder, pass them through zero-convolution layers (Zhang et al., 2023), and feed them into the corresponding upsampling block in the decoder. This modification significantly improves the perceptual similarity between the watermarked image and the input image, as indicated in Table 2. To decode the watermark, we utilize ConvNeXt-B (Liu et al., 2022b) as the decoder, with an additional fully connected layer to output a 100-bit watermark.

### 3.3 OBJECTIVE FUNCTION AND TRAINING STRATEGY

**Objective function.** We follow the standard training scheme, which balances the quality of the watermarked image with the effectiveness of watermark extraction under various image manipulations. The total loss function is as follows:

$$\mathcal{L}_{\text{ALL}} = \mathcal{L}_{\text{IMG}}\left(\boldsymbol{x}_o, \boldsymbol{x}_w\right) + \alpha \mathcal{L}_{\text{BCE}}\left(\boldsymbol{w}, \boldsymbol{w}'\right), \tag{1}$$

where $\alpha$ is a trade-off hyperparameter, $\mathcal{L}_{\text{BCE}}$ is the standard binary cross-entropy loss calculated between the extracted watermark and the ground truth. The image quality loss $\mathcal{L}_{\text{IMG}}$ is defined as:

$$\mathcal{L}_{\text{IMG}} = \beta_{\text{MSE}} \mathcal{L}_{\text{MSE}}\left(\gamma(\boldsymbol{x}_o), \gamma(\boldsymbol{x}_w)\right) + \beta_{\text{LPIPS}} \mathcal{L}_{\text{LPIPS}}\left(\boldsymbol{x}_o, \boldsymbol{x}_w\right) + \beta_{\text{GAN}} \mathcal{L}_{\text{GAN}}\left(\boldsymbol{x}_o, \boldsymbol{x}_w\right), \tag{2}$$

where $\beta_{\text{MSE}}$, $\beta_{\text{LPIPS}}$, and $\beta_{\text{GAN}}$ are the weights of the respective loss terms. Here, $\gamma(\cdot)$ is a differentiable, non-parametric mapping that transforms the input image from RGB color space into the more perceptually uniform YUV color space, $\mathcal{L}_{\text{LPIPS}}\left(\boldsymbol{x}_o, \boldsymbol{x}_w\right)$ is the perceptual loss, and $\mathcal{L}_{\text{GAN}}\left(\boldsymbol{x}_o, \boldsymbol{x}_w\right)$ is a standard adversarial loss from a GAN discriminator $\mathcal{D}_{\text{disc}}$:

$$\mathcal{L}_{\text{GAN}} = \mathbb{E}_{\boldsymbol{x}_o}\left[\log \mathcal{D}_{\text{disc}}(\boldsymbol{x}_o)\right] + \mathbb{E}_{\boldsymbol{x}_o, \boldsymbol{w}}\left[\log\left(1 - \mathcal{D}_{\text{disc}}(E(\boldsymbol{x}_o, \boldsymbol{w}))\right)\right]. \tag{3}$$

**Training strategy.** In the first training stage, we prioritize the watermark extraction loss by setting $\alpha$ to 10 and $\beta_{\text{MSE}}$, $\beta_{\text{LPIPS}}$, $\beta_{\text{GAN}}$ each to 0.01. To preserve the generative prior, the UNet and VAE decoder of the SDXL-Turbo, along with the added zero-convolution layers, are frozen. Once the bit accuracy exceeds 0.85, we transition to the second stage, unfreezing all parameters for further training. At this point, the loss weighting factors are adjusted to $\alpha = 1.5$, $\beta_{\text{MSE}} = 2.0$, $\beta_{\text{LPIPS}} = 1.5$, and $\beta_{\text{GAN}} = 0.5$. The model after the first two stages serves as our base model, dubbed VINE-B. We then fine-tune VINE-B in the third stage by incorporating Instruct-Pix2Pix (Brooks et al., 2023), a representative instruction-driven image editing model, into our noise layer. Gradients are backpropagated through a straight-through estimator (Bengio et al., 2013). Note that it cannot converge if it

is applied directly during the early training stages. The fine-tuned model is referred to as VINE-R. Additional implementation details are provided in Appendix F.

**Resolution scaling.** Different watermarking models are typically trained using a fixed input resolution, limiting them to accepting only fixed-resolution inputs during testing. However, in practical applications, supporting watermarking at the original resolution is crucial to preserve the input image quality. Bui et al. (2023) propose a method (detailed in Appendix D.1) to adapt any watermarking model to handle arbitrary resolutions without compromising the quality of the watermarked images and their inherent robustness, as demonstrated in Appendices D.2 and D.3. In our experiments, we apply this resolution scaling method to all methods, enabling them to operate at a uniform resolution of $512 \times 512$, which is compatible with image editing models.

## 4 EXPERIMENTS

In W-Bench, we assess the robustness of eleven representative watermarking models against a variety of image editing methods, including image regeneration, global editing, local editing, and image-to-video generation. Section 4.1 and Section 4.2 outline the employed image editing methods and the benchmark setup, respectively. Section 4.3 analyzes the benchmarking results. In Section 4.4, we conduct ablation studies to understand the impact of the key components.

### 4.1 IMAGE EDITING METHODS

**Image regeneration.** Image regeneration involves perturbing an image into a noisy version and then reconstructing it. The perturbing process can be either stochastic or deterministic. In the stochastic method (Meng et al., 2021; Nie et al., 2022; Zhao et al., 2023b), random Gaussian noise is introduced to the image, with the noise level typically controlled by a timestep $t_s$ using common noise schedulers such as VP (Ho et al., 2020), VE (Song & Ermon, 2019; Song et al., 2020b), FM (Lipman et al., 2022; Liu et al., 2022a), and EDM (Karras et al., 2022). The diffusion model then denoises the noisy image starting from timestep $t_s$ to produce a clean image. In contrast, the deterministic method (Mokady et al., 2022; Song et al., 2020a; Wallace et al., 2022), also known as image inversion, utilizes a diffusion model to deterministicly invert a clean image into a noisy version through multiple sampling steps $n_d$. The image is then reconstructed by applying the same sampling methods over the same number of sampling steps $n_d$. We employ the widely used VP scheduler for stochastic regeneration, testing noise timesteps $t_s$ from 60 to 240 in increments of 20. For deterministic regeneration, we utilize the fast sampler DPM-solver (Lu et al., 2022a;b) and evaluate sampling steps $n_d$ of 15, 25, 35, and 45. See Appendix G.2 for the effects of image regeneration on images.

**Global and local editing.** Although global editing typically involves stylization, we also consider editing methods guided solely by text prompts. In these cases, unintended background changes frequently occur, regardless of the requested edit—adding, replacing, or removing objects; altering actions; changing colors; modifying text or patterns; or adjusting object quantities. Even though the edited background often appears perceptually similar to the original, these unintended alterations can compromise the embedded watermark. In contrast, local editing refers to editing models that use region masks as input, ensuring that the area outside the mask remains unchanged in the edited image. We employ Instruct-Pix2Pix (Brooks et al., 2023), MagicBrush (Zhang et al., 2024b), and UltraEdit (Zhao et al., 2024c) for global editing, while ControlNet-Inpainting (Zhang et al., 2023) and UltraEdit are used for local editing. Notably, UltraEdit can accept a region mask or operate without it, allowing us to utilize this model for both global and local editing. We use each model's default sampler and perform 50 sampling steps to generate edited images. The difficulty of global editing is controlled by the classifier-free guidance scale of text prompts (Ho & Salimans, 2022), which ranges from 5 to 9, while the image guidance is fixed at 1.5. For local editing, difficulty is determined by the percentage of the edited region with respect to the entire image (i.e., the size of the region mask), with intervals set at 10–20%, 20–30%, 30–40%, 40–50%, and 50–60%. In all cases of local editing, the image and text guidance values are consistently set to 1.5 and 7.5, respectively.

**Image-to-video generation.** In the experiments, we utilize SVD (Blattmann et al., 2023) to generate a video from a single image. We assess whether the watermark remains detectable in the resulting video frames. Since the initial frames closely resemble the input image, we begin our analysis with frame 5 and continue through frame 19, selecting every second frame.

Table 1: Comparison of watermarking performance in terms of watermarked image quality and detection accuracy across various image editing methods at a uniform resolution $512 \times 512$. Quality metrics are averaged over 10,000 images, and the TPR@0.1%FPR for each specific editing method is averaged over 5,000 images. The best value in each column is highlighted in **bold**, and the second best value is underlined. Abbreviations: Cap = Encoding Capacity; Sto = Stochastic Regeneration; Det = Deterministic Regeneration; Pix2Pix = Instruct-Pix2Pix; Ultra = UltraEdit; Magic = MagicBrush; CtrlN = ControlNet-Inpainting; SVD = Stable Video Diffusion.

| Method | Cap↑ | PSNR↑ | SSIM↑ | LPIPS↓ | FID↓ | TPR@0.1%FPR ↑ (%) (averaged over all difficulty levels) | | | | | | | |
| | | | | | | Regeneration | | Global Editing | | | Local Editing | | I2V |
| | | | | | | Sto | Det | Pix2Pix | Ultra | Magic | Ultra | CtrlN | SVD |
| MBRS (Jia et al., 2021) | 30 | 27.37 | 0.8940 | 0.1877 | 6.85 | 99.53 | 99.35 | 83.50 | 7.50 | 88.54 | 99.60 | 89.16 | 13.55 |
| CIN (Ma et al., 2022) | 30 | **43.19** | 0.9847 | 0.0270 | 1.13 | 44.85 | 51.65 | 51.40 | 17.00 | 68.38 | 51.28 | 66.04 | 2.93 |
| PIMoG (Fang et al., 2022) | 30 | 37.72 | 0.9863 | 0.0289 | 3.43 | 82.85 | 71.18 | 72.78 | 40.14 | 81.88 | 74.30 | 64.22 | 14.33 |
| RivaGAN (Zhang et al., 2019) | 32 | 40.43 | 0.9702 | 0.0488 | 1.86 | 10.12 | 12.50 | 6.22 | 4.14 | 33.96 | 34.28 | 56.92 | 3.15 |
| SepMark (Wu et al., 2023) | 30 | 35.48 | 0.9814 | 0.0150 | 1.72 | 61.21 | 73.85 | 87.74 | 51.84 | 82.58 | 92.94 | 97.14 | 8.81 |
| DWTDCT (Al-Haj, 2007) | 30 | 40.46 | 0.9705 | 0.0136 | 0.24 | 0.09 | 0.00 | 0.04 | 0.06 | 0.04 | 0.32 | 0.56 | 0.01 |
| DWTDCTSVD (Navas et al., 2008) | 30 | 40.40 | 0.9799 | 0.0265 | 0.86 | 3.12 | 1.43 | 3.82 | 4.02 | 30.84 | 24.56 | 50.04 | 0.76 |
| SSL (Fernandez et al., 2022) | 30 | 41.77 | 0.9796 | 0.0350 | 3.54 | 1.76 | 9.70 | 25.06 | 10.58 | 50.10 | 25.28 | 31.46 | 3.65 |
| StegaStamp (Tancik et al., 2020) | 100 | 29.65 | 0.9107 | 0.0645 | 7.61 | 91.09 | 92.13 | 93.72 | 51.24 | 91.18 | 98.84 | **99.06** | 30.85 |
| TrustMark (Bui et al., 2023) | 100 | 41.27 | 0.9910 | **0.0026** | 0.86 | 9.22 | 34.20 | 77.72 | 43.48 | 85.90 | 76.62 | 59.78 | **39.60** |
| EditGuard (Zhang et al., 2024d) | 64 | 37.58 | 0.9406 | 0.0171 | 0.51 | 0.09 | 6.00 | 0.06 | 1.16 | 0.24 | 0.18 | 2.66 | 0.18 |
| VINE-Base | 100 | 40.51 | **0.9954** | 0.0029 | **0.08** | 91.03 | 99.25 | 96.30 | 80.90 | 89.29 | 99.60 | 89.68 | 25.44 |
| VINE-Robust | 100 | 37.34 | 0.9934 | 0.0063 | 0.15 | **99.66** | **99.98** | **97.46** | **86.86** | **94.58** | **99.96** | 93.04 | 36.33 |

## 4.2 EXPERIMENTAL SETUP

**Datasets.** We train VINE using the OpenImage dataset (Kuznetsova et al., 2020) at a resolution of $256 \times 256$. The training details are provided in Appendix F. For evaluation, we randomly sample 10,000 instances from the UltraEdit dataset (Zhao et al., 2024c), each containing a source image, an editing prompt, and a region mask. The images in UltraEdit dataset are photographs sourced from datasets such as COCO (Lin et al., 2014), Flickr (Young et al., 2014), and ShareGPT4V (Chen et al., 2023). Of these 10,000 samples, 1,000 are allocated for stochastic regeneration, another 1,000 for deterministic regeneration, and 1,000 for global editing. For local editing, 5,000 samples are designated for local editing. These consist of five sets, each containing 1,000 images, which are edited using mask sizes covering 10–20%, 20–30%, 30–40%, 40–50%, and 50–60% of the total image area. Additionally, we include 1,000 samples for image-to-video generation and 1,000 for testing conventional distortion, thereby completing the total evaluation set.

**Baselines.** We compare VINE with eleven watermark baselines, all utilizing their officially released checkpoints. These baselines include MBRS (Jia et al., 2021), CIN (Ma et al., 2022), PIMoG (Fang et al., 2022), RivaGAN (Zhang et al., 2019), SepMark (Wu et al., 2023), TrustMark (Bui et al., 2023), DWTDCT (Al-Haj, 2007), DWTDCTSVD (Navas et al., 2008), SSL (Fernandez et al., 2022), StegaStamp (Tancik et al., 2020), and EditGuard (Zhang et al., 2024d). Although the baselines were trained at different fixed resolutions (as detailed in Appendix D), we apply resolution scaling (described in Section 3.3) to standardize all of them to a uniform resolution of $512 \times 512$. This standardization does not compromise their robustness, as demonstrated in Appendix D.3.

**Metrics.** We evaluate the imperceptibility of watermarking models using standard metrics, including PSNR, SSIM, LPIPS (Zhang et al., 2018), and FID (Parmar et al., 2022). For watermark extraction, it is essential to strictly control the false positive rate (FPR), as incorrectly labeling non-watermarked images as watermarked can be detrimental—a concern often overlooked in previous studies. *Neither high bit accuracy nor AUROC alone guarantees a high true positive rate (TPR) at a low FPR.* A detailed discussion is provided in Appendix C. Thus, we primarily focus on TPR@0.1%FPR and TPR@1%FPR as our main metrics. Accordingly, both the watermarked and original images are fed into watermark decoders for evaluation. Additionally, we also provide bit accuracy and AUROC for reference. Note that all reported baseline bit accuracies do not include error correction methods, such as BCH (Bose & Ray-Chaudhuri, 1960), which can be applied to all watermarking models.

## 4.3 BENCHMARKING RESULTS AND ANALYSIS

Table 1 summarizes the overall evaluation results. As discussed in Section 4.2, we report TPR@0.1%FPR as the primary metric, with additional metrics provided in Figure 5. Each reported TPR@0.1%FPR value is averaged across a total of $m \times 1,000$ images, where $m$ represents the number of difficulty levels for the specific image editing task. The quality metrics—PSNR, SSIM, LPIPS, and FID—are calculated for each pair of watermarked and input images and then averaged across

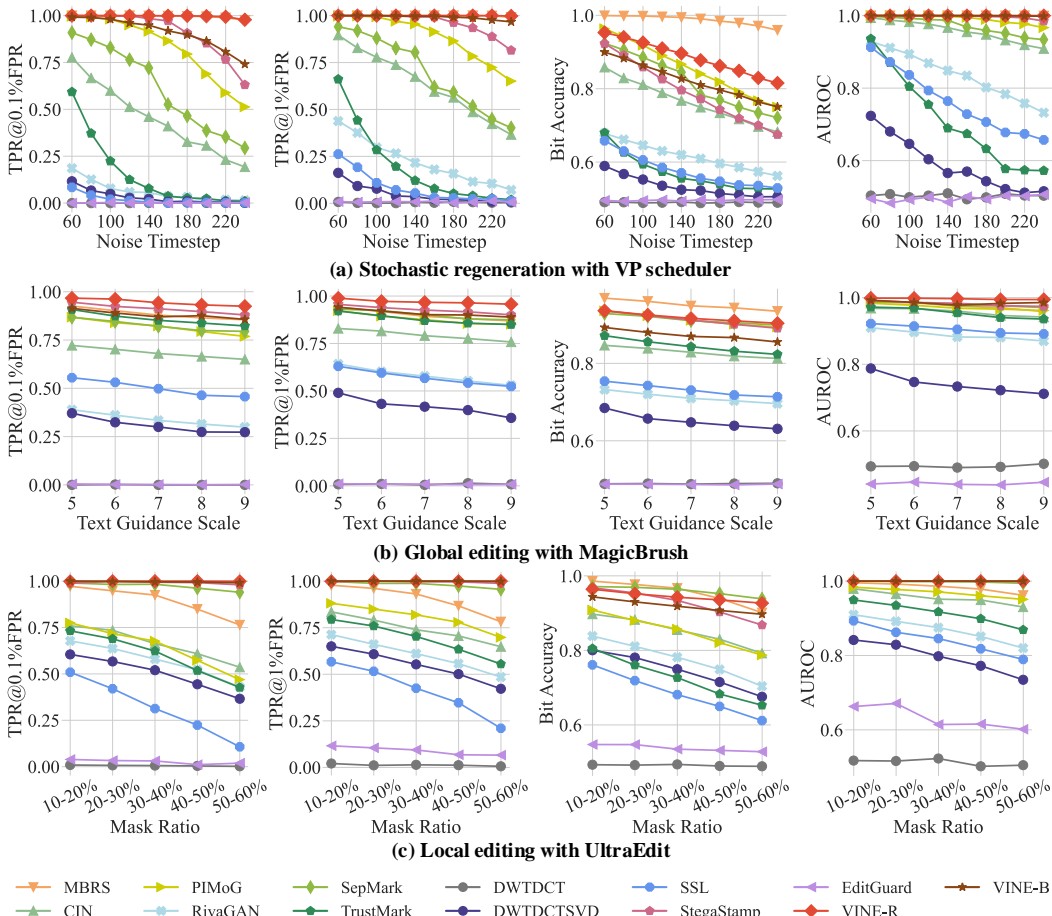

Figure 5: The performance of watermarking methods under (a) Stochastic regeneration, (b) Global editing, and (c) Local editing. Additional results are available in Figure 18.

all 10,000 image pairs. MBRS and StegaStamp perform well in image regeneration and local editing tasks; however, they have lower image quality. Qualitative comparison is provided in Appendix I. Additionally, MBRS has a limited encoding capacity of only 30 bits. Although SepMark, PIMoG, and TrustMark strike a better balance between image quality and detection accuracy, their detection accuracy remains unsatisfactory. In contrast, our methods, VINE-B and VINE-R, offer the best trade-off. VINE-B provides superior image quality with slightly lower detection accuracy under image editing, while VINE-R delivers greater robustness by sacrificing some image quality. EditGuard is not designed for robust watermarking against image editing, as it is trained with mild degradation. Instead, it offers a feature for tamper localization, enabling the identification of edited regions.

Figure 5 illustrates the watermarking performance across various difficulty levels for different image editing methods. The evaluation results against representative editing models are displayed, while additional results are available in Appendix J. For image regeneration, both VINE-R and MBRS maintain high TPR@0.1%FPR across various difficulty levels. VINE-B, StegaStamp, and PIMoG perform well when subjected to minor perturbations. It is important to note that stochastic regeneration with large noise perturbation steps can significantly alter the image, as shown in Appendix G.2. Although SSL achieves higher bit accuracy and AUROC than TrustMark, it exhibits a lower TPR@0.1%FPR. Further investigation revealed that SSL has a high FPR, often producing bit accuracy exceeding 0.7 for unwatermarked images. Therefore, bit accuracy and AUROC alone are insufficient for evaluating watermarking performance.

For global editing and local editing, VINE-B, VINE-R, and StegaStamp achieve high TPR@0.1%FPR across various classifier-free guidance scales. Notably, we also use UltraEdit for global editing, which provides better alignment between the edited image and the editing instructions compared to instruct-Pix2Pix and MagicBrush (a quantitative analysis of editing models is presented in Appendix G.3). However, methods that perform well for local editing with UltraEdit

Table 2: Ablation study examining the impact of key components on image regeneration and global editing. Each configuration builds upon the previous one, with changes highlighted in red.

| Config | Blurring Distortions | Watermark Encoder | | | | | PSNR ↑ | SSIM ↑ | LPIPS ↓ | FID ↓ | TPR@0.1%FPR ↑ (%) | | | |
|---|---|---|---|---|---|---|---|---|---|---|---|---|---|---|
| | | Backbone | Condition | Skip | Pretrained | Finetune | | | | | Sto | Det | Pix2Pix | Ultra |
| Config A | ✘ | Simple UNet | N.A. | N.A. | N.A. | ✘ | 38.21 | 0.9828 | 0.0148 | 1.69 | 54.61 | 66.86 | 64.24 | 32.62 |
| Config B | ✔ | | | | | ✘ | 35.85 | 0.9766 | 0.0257 | 2.12 | 86.85 | 92.28 | 80.98 | 62.14 |
| Config C | ✔ | | | | | ✔ | 31.24 | 0.9501 | 0.0458 | 4.67 | 98.59 | 99.29 | 96.01 | 84.60 |
| Config D | ✔ | SDXL-Turbo | ControlNet | ✘ | ✔ | ✘ | 32.68 | 0.9640 | 0.0298 | 2.87 | 90.82 | 94.89 | 91.86 | 70.69 |
| Config E | ✔ | | Cond. Adaptor | ✘ | ✔ | ✘ | 36.76 | 0.9856 | 0.0102 | 0.53 | 90.86 | 94.78 | 92.88 | 70.68 |
| Config F (VINE-B) | ✔ | | Cond. Adaptor | ✔ | ✔ | ✘ | **40.51** | **0.9954** | **0.0029** | **0.08** | 91.03 | 99.25 | 96.30 | 80.90 |
| Config G (VINE-R) | ✔ | | Cond. Adaptor | ✔ | ✔ | ✔ | 37.34 | 0.9934 | 0.0063 | 0.15 | **99.66** | **99.98** | **97.46** | **86.86** |
| Config H | ✔ | | Cond. Adaptor | ✔ | ✘ | ✔ | 35.18 | 0.9812 | 0.0137 | 1.03 | 99.67 | 99.92 | 96.13 | 84.66 |

do not perform satisfactorily when applied to global editing using the same model, as shown in Figure 18(b). This suggests that global editing more significantly degrades watermarks. Image-to-video generation is not a form of traditional image editing, but we are interested in whether watermarks can persist in the generated frames. As illustrated in Figure 18(e), the overall detection rate is not high. Upon analyzing I2V generation in the frequency domain, we discovered that this process significantly reduces the intensity of nearly all patterns across all frequency bands, rendering them unobservable to the human eye. We infer this is because the generated video frames alter the image layout as objects or the camera move. In this case, the intensity of the watermarking patterns should be substantially increased, at least to levels exceeding those shown in Figure 7.

## 4.4 ABLATION STUDY

In this section, we showcase the effectiveness of our designs through an extensive ablation study summarized in Table 2. We start with **Config A**, a baseline utilizing a simple UNet as the watermark encoder and incorporating only the common distortions outlined in Section 3.1. Building upon this, **Config B** introduces blurring distortions to the noise layer, which significantly enhances robustness against image editing but compromises image quality. **Config C** further refines Config B by using a straight-through estimator to fine-tune with Instruct-Pix2Pix. This enhances robustness, albeit at the cost of some image quality. **Config D** replaces the UNet backbone with the pretrained SDXL-Turbo and integrates image and watermark conditions via ControlNet (Zhang et al., 2023), boosting robustness while degrading image quality due to conflicts from the additional branch. **Config E** substitutes ControlNet with our condition adaptor, restoring image quality to the level comparable with Config B while maintaining the robustness of Config D. **Config F (VINE-B)** enhances Config E by introducing skip connections and zero-convolution layers, further improving both image quality and robustness. **Config G (VINE-R)** fine-tunes Config F using a straight-through estimator with Instruct-Pix2Pix, which increases robustness but reduces image quality. Notably, compared to Config C, Config G leverages a larger model and a powerful generative prior, resulting in significant improvements in image quality and modest gains in robustness. Finally, **Config H** is trained with randomly initialized weights instead of pretrained ones while retaining all other settings from Config G, leading to lower image quality (particularly on the FID metric) but no change in robustness.

## 5 CONCLUSION

In this work, we introduce W-Bench, the first comprehensive benchmark that incorporates four types of image editing powered by large-scale generative models to evaluate the robustness of watermarking models. Eleven representative watermarking methods are selected and tested on W-Bench. We demonstrate how image editing commonly affects the Fourier spectrum of the image and identify an effective and efficient surrogate to simulate these effects during training. Our model, VINE, achieves outstanding watermarking performance against various image editing techniques, outperforming prior methods in both image quality and robustness. These results suggest that one-step pre-trained models can serve as strong and versatile backbones for watermarking, and that a powerful generative prior enhances information embedding in a more imperceptible and robust manner.

**Limitations.** While our method delivers exceptional performance against common image editing tasks powered by generative models, its effectiveness in I2V generation remains limited. Moreover, our model is larger than the baseline models, leading to increased memory requirements and slightly slower inference speeds, as detailed in Table 7.

ACKNOWLEDGMENTS

This research is supported by the National Research Foundation, Singapore and Infocomm Media Development Authority under its Trust Tech Funding Initiative. Any opinions, findings and conclusions or recommendations expressed in this material are those of the author(s) and do not reflect the views of National Research Foundation, Singapore and Infocomm Media Development Authority.

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

# Appendix

## Table of Contents

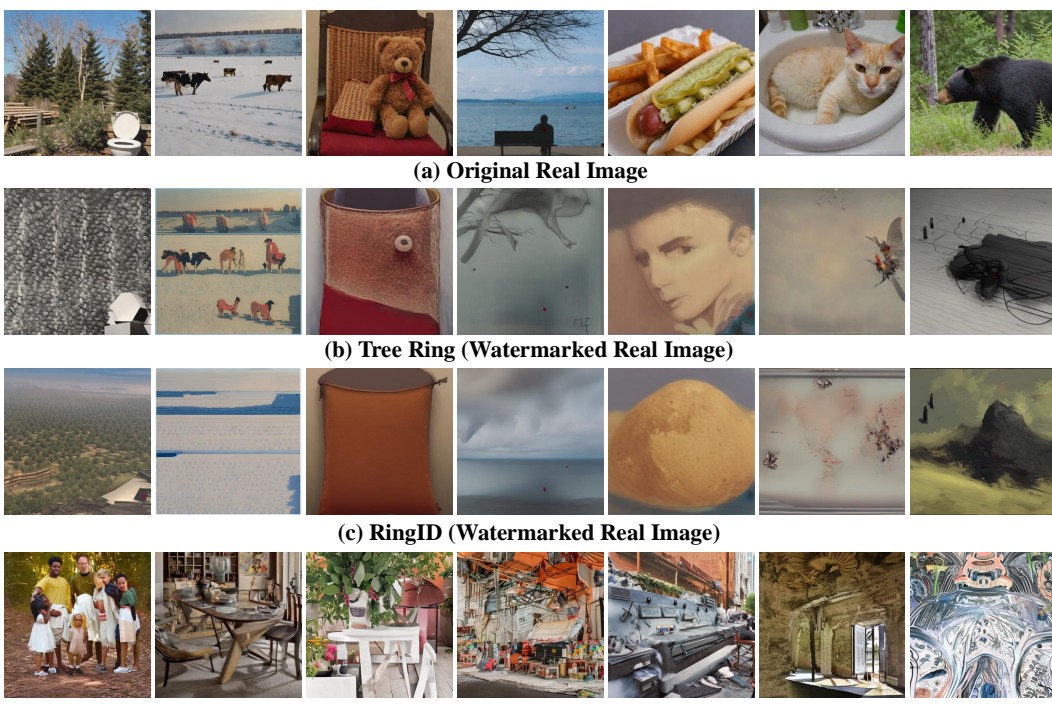

(a) Original Real Image

(b) Tree Ring (Watermarked Real Image)

(c) RingID (Watermarked Real Image)

(d) Gaussian Shading (Watermarked Real Image)

Figure 6: Comparison of Tree Ring (Wen et al., 2023), RingID (Ci et al., 2024), and Gaussian Shading (Yang et al., 2024) methods for embedding watermarks in **real images**. Each real image is first inverted into noise using a 50-step image inversion process before the watermark is injected.

## A  RELATED WORK

### A.1  IN-GENERATION IMAGE WATERMARKING

Another line of research involves watermarking generated images by altering the generation process or fine-tuning the generative models. These methods (Cui et al., 2023; Ci et al., 2024; Fernandez et al., 2023; Liu et al., 2023; Meng et al., 2024; Rezaei et al., 2024; Wen et al., 2023; Xiong et al., 2023; Yang et al., 2024; Zhao et al., 2023c; Zhang et al., 2024a), also known as in-generation image watermarking, aim to facilitate the detection of AI-generated content by inherently embedding a watermark during image generation. These methods can also help protect the copyright of the generated content. However, since in-generation watermarking can only be applied to generated images, not real ones, and this study focuses on protecting copyright and intellectual property in real-world scenarios, these techniques are not included in our benchmark. Nonetheless, it is important to note that our proposed methods could also enhance the robustness of in-generation watermarks against image editing.

Figure 6 shows the results of applying in-generation watermarking methods combined with image inversion techniques to embed watermarks in real images. Specifically, we applied DDIM/DPM inversion methods to extract initial noises from real images, added watermarks using three in-generation watermarking techniques—Tree Ring (Wen et al., 2023), RingID (Ci et al., 2024), and Gaussian Shading (Yang et al., 2024)—and then inverted the noise back to produce the watermarked image. The resulting image differed significantly from the original, thereby undermining the photographer's intent to protect their work.

### A.2  GENERATIVE MODEL-BASED IMAGE EDITING

Recent and significant advancements in text-to-image generative models (Chang et al., 2023; Ding et al., 2022; Nichol et al., 2021; Ramesh et al., 2022; Rombach et al., 2022; Saharia et al., 2022; Yu et al., 2022; Peebles & Xie, 2023; Esser et al., 2024) have enhanced numerous applications (Avrahami et al., 2023; Ruiz et al., 2023; Hertz et al., 2022; Kim et al., 2022; Tumanyan et al., 2023;

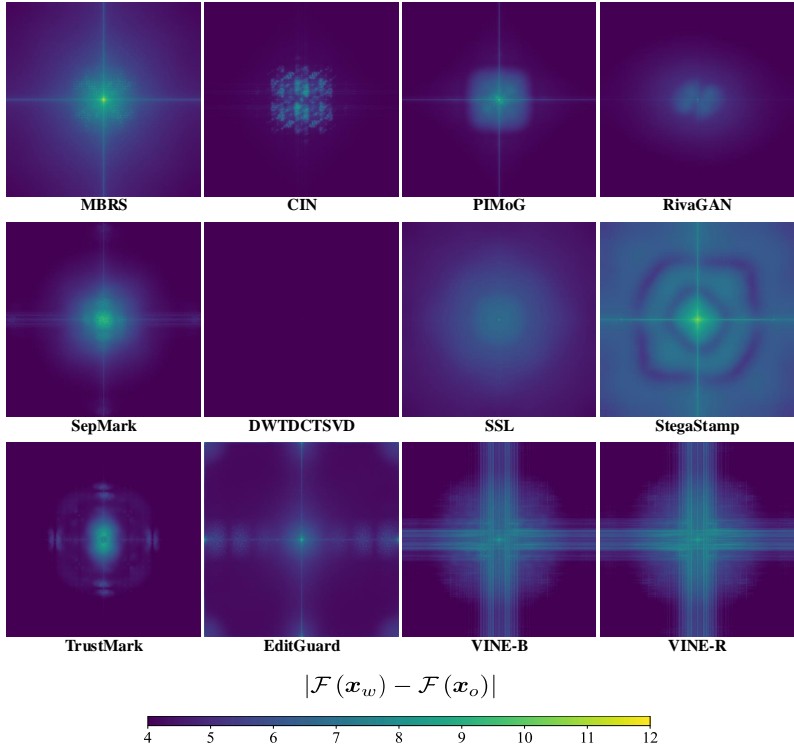

$$|\mathcal{F}\left(\boldsymbol{x}_{w}\right) - \mathcal{F}\left(\boldsymbol{x}_{o}\right)|$$

Figure 7: Frequency pattern visualizations for each watermarking method. Each subfigure displays the absolute difference between the Fourier spectrum $\mathcal{F}(\cdot)$ of the watermarked image $\boldsymbol{x}_w$ and the original image $\boldsymbol{x}_o$, i.e., $|\mathcal{F}(\boldsymbol{x}_w) - \mathcal{F}(\boldsymbol{x}_o)|$. The DWTDCT method is excluded because it closely resembles DWTDCTSVD and their pattern intensity is too weak to be discerned on the uniform scale. Please zoom in for a closer look.

Zhang et al., 2023; Mou et al., 2023; Shi et al., 2024; Zhu et al., 2023; Zhou et al., 2023; 2024a; Lu et al., 2023; 2024; Zhao et al., 2023a; 2024a;b; Wang & Chen, 2024; Gandikota et al., 2023; Parmar et al., 2024; Zhou et al., 2024b;c; Zeng et al., 2024; Peng et al., 2024b;a;c; Gao et al., 2024; Zhu et al., 2024). In this study, we focus on real image editing, which allows users to freely modify actual photographs, producing highly realistic results. Typically, the inputs for image editing include an image and various conditions that help users accurately describe their desired changes. These conditions can encompass text prompts using natural language to specify the edits (Brooks et al., 2023; Zhang et al., 2024c; Fu et al., 2023; Zhang et al., 2024b), region masks to designate areas for modification (Zhao et al., 2024c; Zhuang et al., 2023; Wang et al., 2023), additional images to provide desired styles or objects (Chen et al., 2024; Lu et al., 2023; Yang et al., 2023), and drag points (Pan et al., 2023; Shi et al., 2024; Mou et al., 2023) that enable users to interactively move specific points in the image to target positions.

We broadly categorize image editing into two types: local editing and global editing. Local editing involves modifying only a specific region of an image while keeping the other areas unchanged. In contrast, global editing modifies the entire image, though certain changes may occur unintentionally due to the nature of generative models, rather than users' intent. These editing methods can compromise embedded watermarks, potentially undermining copyright protection. Note that we do not include drag-based editing methods (Pan et al., 2023; Shi et al., 2024; Mou et al., 2023) in our benchmark. This exclusion is due to the lengthy optimization times required to edit a single image using these methods and the limited availability of datasets that provide valid drag points. Extending the benchmark to include drag-based methods presents a potential direction for future research.

## B  WATERMARKING FREQUENCY PATTERNS

In this analysis, we examine each watermarking method within the frequency domain to understand how they alter the frequency spectrum of input images. Specifically, we calculate the absolute dif-

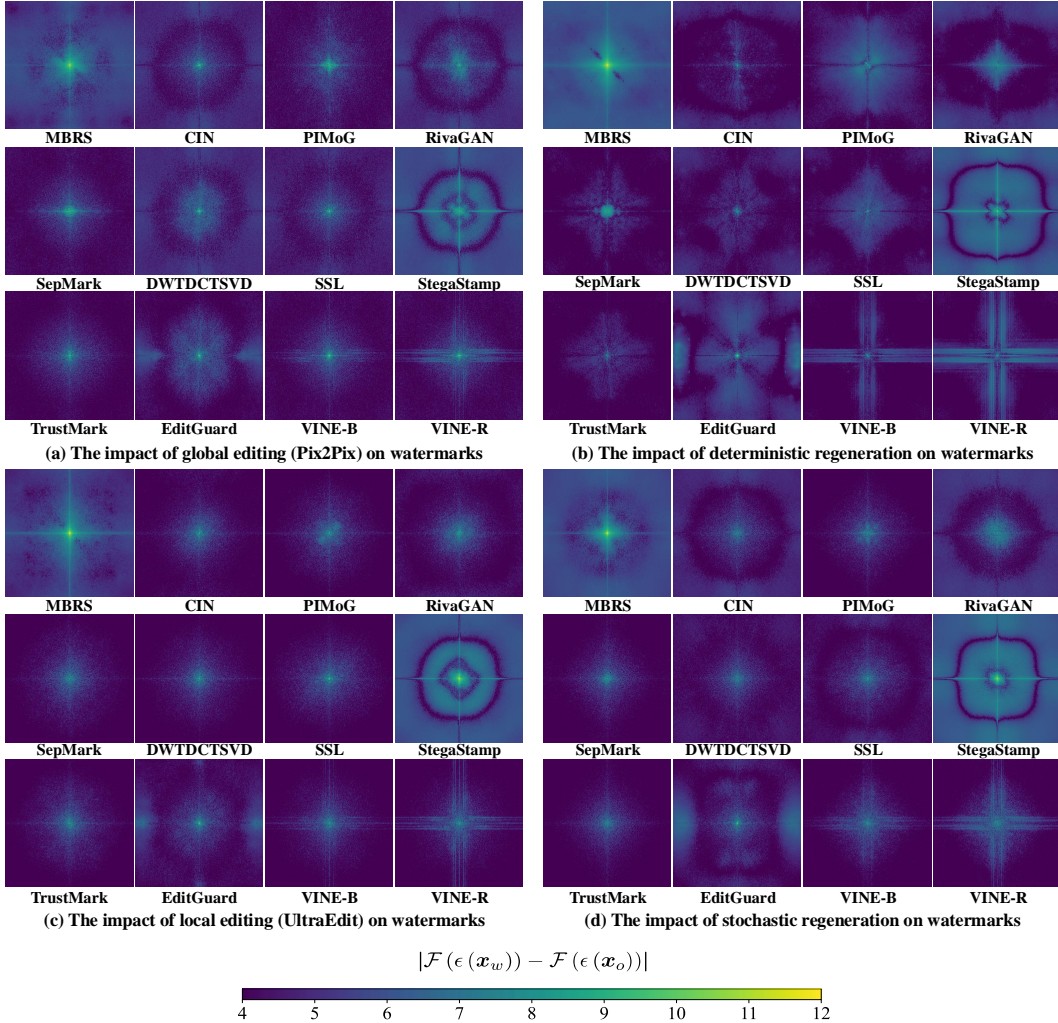

$$|\mathcal{F}\left(\epsilon\left(\boldsymbol{x}_w\right)\right) - \mathcal{F}\left(\epsilon\left(\boldsymbol{x}_o\right)\right)|$$

Figure 8: Impact of different editing methods on the frequency patterns of various watermarks. Each subfigure, analogous to Figure 3, displays the absolute difference between the Fourier spectrum $\mathcal{F}(\cdot)$ of the edited watermarked image $\epsilon(\boldsymbol{x}_w)$ and the original image $\epsilon(\boldsymbol{x}_o)$, i.e., $|\mathcal{F}(\epsilon(\boldsymbol{x}_w)) - \mathcal{F}(\epsilon(\boldsymbol{x}_o))|$ to evaluate how the watermark patterns are altered by the editing process. The frequency patterns of VINE-R, VINE-B, MBRS, and StegaStamp are less affected compared to their original patterns (shown in Figure 7) than those of other watermarking methods. Please zoom in for a closer look.

ference between the Fourier spectrum $\mathcal{F}(\cdot)$ of the watermarked image $\boldsymbol{x}_w$ and the original image $\boldsymbol{x}_o$ as $|\mathcal{F}(\boldsymbol{x}_w) - \mathcal{F}(\boldsymbol{x}_o)|$, and then average this metric over 1,000 pairs of watermarked and original images. Figure 7 presents the results on a logarithmic scale.

Interestingly, aside from our method, **the top four watermarking methods that demonstrate robustness against image editing in certain scenarios (as illustrated in Figure 1)—namely StegaStamp, MBRS, SepMark, and PIMoG—all exhibit prominent patterns in the low-frequency bands, accompanied by a cross-shaped high-intensity pattern.** Among these, StegaStamp shows the strongest pattern intensity. This cross pattern is also part of the low-frequency spectrum, as the Y-axis represents zero frequency in the X-direction, and the X-axis represents zero frequency in the Y-direction. Although CIN and TrustMark also focus their influence on the low-frequency bands, their robustness remains relatively low. We infer that this may be related to the absence of the cross-shaped pattern and the lower intensity of their patterns.

**These patterns support our conclusion in Section 3.1: the more a watermark's impact is concentrated in the low-frequency bands of the spectrum, the more robust it is against image editing.** Although the model trained using our method, VINE, exhibits less pronounced influence along the X and Y axes compared to StegaStamp, it displays highly dense patterns near these axes, which we infer contribute to its enhanced robustness. Additionally, compared to VINE-B, VINE-R shows higher brightness in the central region, further increasing its robustness.

Figure 8 demonstrates the effect of various editing methods on different watermark patterns within the frequency domain. As illustrated, the frequency patterns of VINE-R, VINE-B, MBRS, and StegaStamp are less affected compared to their original patterns (shown in Figure 7) than those of other watermarking methods. However, it challenging to isolate the effects on low-, mid-, and high-frequency bands when directly observing the watermarks.

Note that this finding is a byproduct of our design rather than a deliberate motivation of it. Our two key design elements (surrogate attack & generative prior adaptation) are based on the observations: (1) Image editing and image blurring exhibit similar frequency characteristics. (2) Image watermarking can be viewed as a form of conditional generation, where a generative prior can enhance image quality by making watermarks less visible. Regarding the intriguing finding—the robustness of our watermarking model against image editing being highly positively correlated with pattern intensity in the low-frequency region—this emerged from our training process rather than driving our design decisions. This finding aligns well with our results in Section 3.1, which show that image editing affects patterns in the high-frequency region more than those in the low-frequency region.

## C  STATISTICAL TEST

Let $\boldsymbol{w} \in \{0,1\}^k$ be the $k$-bit ground-truth watermark. The watermark $\boldsymbol{w}'$ extracted from a watermarked image $\boldsymbol{x}_w$ is compared with the ground-truth $\boldsymbol{w}$ for detection. The detection statistical test relies on the number of matching bits $M(\boldsymbol{w}, \boldsymbol{w}')$: If

$$M(\boldsymbol{w}, \boldsymbol{w}') \geq \tau, \quad \text{where } \tau \in \{0, 1, 2, \cdots, k\}, \tag{4}$$

then the image is flagged as watermarked. Formally, we test the statistical hypothesis $H_1$: '$\boldsymbol{x}$ contains the watermark $\boldsymbol{w}$' against the null hypothesis $H_0$: '$\boldsymbol{x}$ does not contain the watermark $\boldsymbol{w}$'. Under $H_0$ (i.e., for original images), if the extracted bits $\boldsymbol{w} = \{w_1', w_2', \cdots, w_k'\}$ (where $w_i'$ is the i-th extracted bit) from a model are independent and identically distributed (i.i.d.) Bernoulli random variables with the matching probability $p_o$, then $M(\boldsymbol{w}, \boldsymbol{w}')$ follows a binomial distribution with parameters $(k, p_o)$. This assumption is verified by Fernandez et al. (2023).

The false positive rate (FPR) is the probability that $M(\boldsymbol{w}, \boldsymbol{w}')$ takes a value bigger than the threshold $\tau$ under the null hypothesis $H_0$. It is obtained from the CDF of the binomial distribution, and a closed-form can be written with the regularized incomplete beta function $I_p(a, b)$ (Fernandez et al., 2023):

$$\text{FPR}(\tau) = \mathbb{P}\left(M(\boldsymbol{w}, \boldsymbol{w}') > \tau | H_0\right) = \sum_{i=\tau+1}^{k} \binom{k}{i} p_o^i (1 - p_o)^{k-i} = I_{p_o}(\tau + 1, k - \tau), \tag{5}$$

where under $H_0$ (i.e., images without the watermark $\boldsymbol{w}$), $p_o$ should ideally be close to 0.5 to minimize the risk of false positive detection.

Similarly, the true positive rate (TPR) represents the probability that the number of matching bits exceeds the threshold $\tau$ under $H_1$, where the image contains the watermark. Thus, the TPR can be calculated by:

$$\text{TPR}(\tau) = \mathbb{P}\left(M(\boldsymbol{w}, \boldsymbol{w}') > \tau | H_1\right) = \sum_{i=\tau+1}^{k} \binom{k}{i} p_w^i (1 - p_w)^{k-i} = I_{p_w}(\tau + 1, k - \tau), \tag{6}$$

where under $H_1$ (i.e., images with the watermark $\boldsymbol{w}$), $p_w$ should ideally be high enough (e.g., exceeding 0.8) to ensure the effectiveness of a watermarking model.

To further demonstrate that *neither high bit accuracy nor AUROC alone guarantees a high TPR at a low FPR*, consider the following example. Suppose we have a 100-bit watermarking model with a threshold $\tau$ of 70 to determine whether an image contains watermark $\boldsymbol{w}$. If the model extracts bits from watermarked images with a matching probability $p_w = 0.8$ and from original images with a matching probability $p_o = 0.5$, the resulting FPR would be $1.6 \times 10^{-5}$ and the TPR would be 0.99. In this scenario, even though the bit accuracy for watermarked images is not exceptionally high (e.g., below 0.9), the model still achieves a high TPR at a very low FPR. In contrast, if another model has $p_w = 0.9$ and $p_o = 0.7$, achieving the same FPR would require setting the threshold $\tau$ to 87. Under these conditions, the TPR would only be 0.8. This example demonstrates that high bit

accuracy for watermarked images does not necessarily ensure a high TPR when maintaining a low FPR. Therefore, relying solely on bit accuracy or AUROC may not be sufficient for achieving the desired performance in watermark detection.

## D RESOLUTION SCALING

### D.1 METHOD

Bui et al. (2023) propose a method to adapt any watermarking model to handle arbitrary resolutions, as presented in Algorithm 1. This approach preserves the quality of the watermarked images and maintains or even improves their robustness against the image transformations that the models can inherently handle at their native resolution. In our experiments, we apply this resolution scaling method to all methods, enabling them to operate at a uniform resolution of 512×512, which is compatible with image editing models.

---

**Algorithm 1** Resolution scaling

---

1: **Input:** Input image $x_o$, binary watermark $w$
2: **Output:** Watermarked image $x_w$
3: **Model:** Watermark Encoder $E(\cdot)$ trained on the resolution of $u \times v$

4: $h, w \leftarrow \texttt{Size}(x_o)$
5: $x_o \leftarrow x_o/127.5 - 1$ // normalize to range [-1, 1]
6: $x'_o \leftarrow \texttt{interpolate}(x_o, (u, v))$
7: $r' \leftarrow E(x'_o) - x'_o$ // residual image
8: $r \leftarrow \texttt{interpolate}(r', (h, w))$
9: $x_w \leftarrow \texttt{clamp}(x_o + r, -1, 1)$
10: $x_w \leftarrow x_w \times 127.5 + 127.5$

---

### D.2 IMPACT ON WATERMARKED IMAGE QUALITY

Among the evaluated methods, MBRS (Jia et al., 2021), CIN (Ma et al., 2022), PIMoG (Fang et al., 2022), SepMark (Wu et al., 2023), StegaStamp (Tancik et al., 2020), TrustMark (Bui et al., 2023), VINE-B, and VINE-R are trained at resolutions lower than 512×512, necessitating resolution scaling during the encoding process. Table 3 presents the watermarked image quality at their original training resolutions, enabling a comparison with their performance after resolution scaling. It is observed that, in terms of image quality, most methods show a slight improvement in PSNR, SSIM, and FID after resolution scaling, while exhibiting a slight increase in LPIPS.

Table 3: Comparison of watermarking performance, evaluating both image quality of the watermarked images and detection accuracy under normal conditions (no distortion or editing applied) at the original training resolution. The best value in each column is highlighted in **bold**, and the second best value is underlined.

| Method | Resolution | Capacity ↑ | PSNR ↑ | SSIM ↑ | LPIPS ↓ | FID ↓ | TPR@0.1%FPR ↑ (%) |
|---|---|---|---|---|---|---|---|
| MBRS (Jia et al., 2021) | $128 \times 128$ | 30 | 25.14 | 0.8348 | 0.0821 | 13.51 | 100.0 |
| CIN (Ma et al., 2022) | $128 \times 128$ | 30 | **41.70** | 0.9812 | **0.0011** | 2.20 | 100.0 |
| PIMoG (Fang et al., 2022) | $128 \times 128$ | 30 | 37.54 | 0.9814 | 0.0140 | 2.97 | 100.0 |
| SepMark (Wu et al., 2023) | $128 \times 128$ | 30 | 35.50 | 0.9648 | 0.0116 | 2.95 | 100.0 |
| StegaStamp (Tancik et al., 2020) | $400 \times 400$ | 100 | 29.33 | 0.8992 | 0.1018 | 8.29 | 100.0 |
| TrustMark (Bui et al., 2023) | $256 \times 256$ | 100 | 40.94 | 0.9819 | 0.0015 | 1.04 | 100.0 |
| VINE-Base | $256 \times 256$ | 100 | 40.22 | **0.9961** | 0.0022 | **0.10** | 100.0 |
| VINE-Robust | $256 \times 256$ | 100 | 37.07 | 0.9942 | 0.0048 | 0.19 | 100.0 |

### D.3 IMPACT ON ROBUSTNESS AGAINST DISTORTIONS

Figure 9 demonstrates the robustness of the evaluated watermarking methods against classical transmission distortions at a resolution of 512×512 pixels. In contrast, Figure 10 examines watermarking methods—including MBRS, CIN, PIMoG, SepMark, StegaStamp, TrustMark, VINE-B, and VINE-R—that were originally trained at resolutions lower than 512×512 pixels by evaluating them at their

respective training resolutions. This comparison helps determine whether scaling the resolution affects their inherent robustness.

**The results indicate that inherent robustness is either unaffected or even enhanced.** This occurs because, for the same level of distortion, larger images experience comparatively less impact. This phenomenon is particularly evident with Gaussian blurring. At a resolution of $512 \times 512$ pixels, MBRS, PIMoG, SepMark, CIN, and TrustMark can withstand Gaussian blurring with a kernel size of 5 (see Figure 9(a)). However, at their original resolutions of $128 \times 128$ or $256 \times 256$ pixels, a kernel size of 5 has a more significant impact compared to the $512 \times 512$ resolution, resulting in poorer performance (see Figure 10(a)). It is important to note that the kernel size range on the x-axis in Figure 10 is halved compared to Figure 9 because the image resolution is reduced by half or more, resulting in larger kernel sizes that cause excessive blurring.

Although comparing different methods at their original training resolutions is somewhat unfair due to their varying training resolutions, the results demonstrate that resolution scaling does not compromise the robustness of these watermarking methods and can sometimes even enhance it. For other types of attacks, the difficulty levels remain consistent across different resolutions. Additionally, scaling the resolution significantly improves robustness against JPEG compression while maintaining robustness against brightness adjustments, contrast modifications, and Gaussian noise.

Another noteworthy observation is that, in Figure 9(a) (i.e., at the $512 \times 512$ resolution), **the methods that exhibit strong robustness against Gaussian blurring also demonstrate robustness against image editing.** This finding corroborates our conclusions in Section 3.1, as the characteristics of image editing in the frequency domain are similar to those of blurring distortions.

# E    CONDITION ADAPTER ARCHITECTURE

Figure 11 showcases the condition adaptor architecture within our watermark encoder (Figure 4). This architecture consists of multiple fully connected and convolutional layers, each followed by a ReLU activation layer.

# F    IMPLEMENTATION DETAILS

## F.1    PRE-TRAINING

The complete training objective is detailed in Section 3.3. To train VINE-B, we initially prioritize the watermark extraction loss by setting $\alpha$ to 10 and each of $\beta_{\text{MSE}}$, $\beta_{\text{LPIPS}}$, and $\beta_{\text{GAN}}$ to 0.01. To preserve the generative prior, the UNet and VAE decoder of the SDXL-Turbo, along with the additional zero-convolution layers, are kept frozen. Once the bit accuracy exceeds 0.85, we proceed to the second stage by unfreezing all parameters for further training. At this stage, the loss weighting factors are adjusted to $\alpha = 1.5$, $\beta_{\text{MSE}} = 2.0$, $\beta_{\text{LPIPS}} = 1.5$, and $\beta_{\text{GAN}} = 0.5$. In both stages, we employ the Adam optimizer with a learning rate of $1 \times 10^{-4}$ and a batch size of 112, training on the entire OpenImage dataset (Kuznetsova et al., 2020) at a resolution of $256 \times 256$ pixels. We apply transformations that randomly select either random cropping or resizing, followed by center cropping. VINE-B is trained on $8 \times$NVIDIA A100-80GB GPUs for 111k iterations.

## F.2    FINE-TUNING

VINE-R is obtained by further fine-tuning VINE-B on the Instruct-Pix2Pix dataset (Brooks et al., 2023). During this fine-tuning stage, we reduce the learning rate to 5e-6 and adjust the batch size to 80, while keeping the weighting factors for different loss terms the same as in the second stage. Watermarked images are edited using Instruct-Pix2Pix with text prompts and 25 denoising steps. Although the quality of images edited with 25 steps is not as high as those edited with 50 steps, it remains satisfactory and effectively doubles the fine-tuning speed. Gradients are then backpropagated using a straight-through estimator (Bengio et al., 2013). VINE-R is fine-tuned using $8 \times$NVIDIA A100-80GB over 80k iterations.

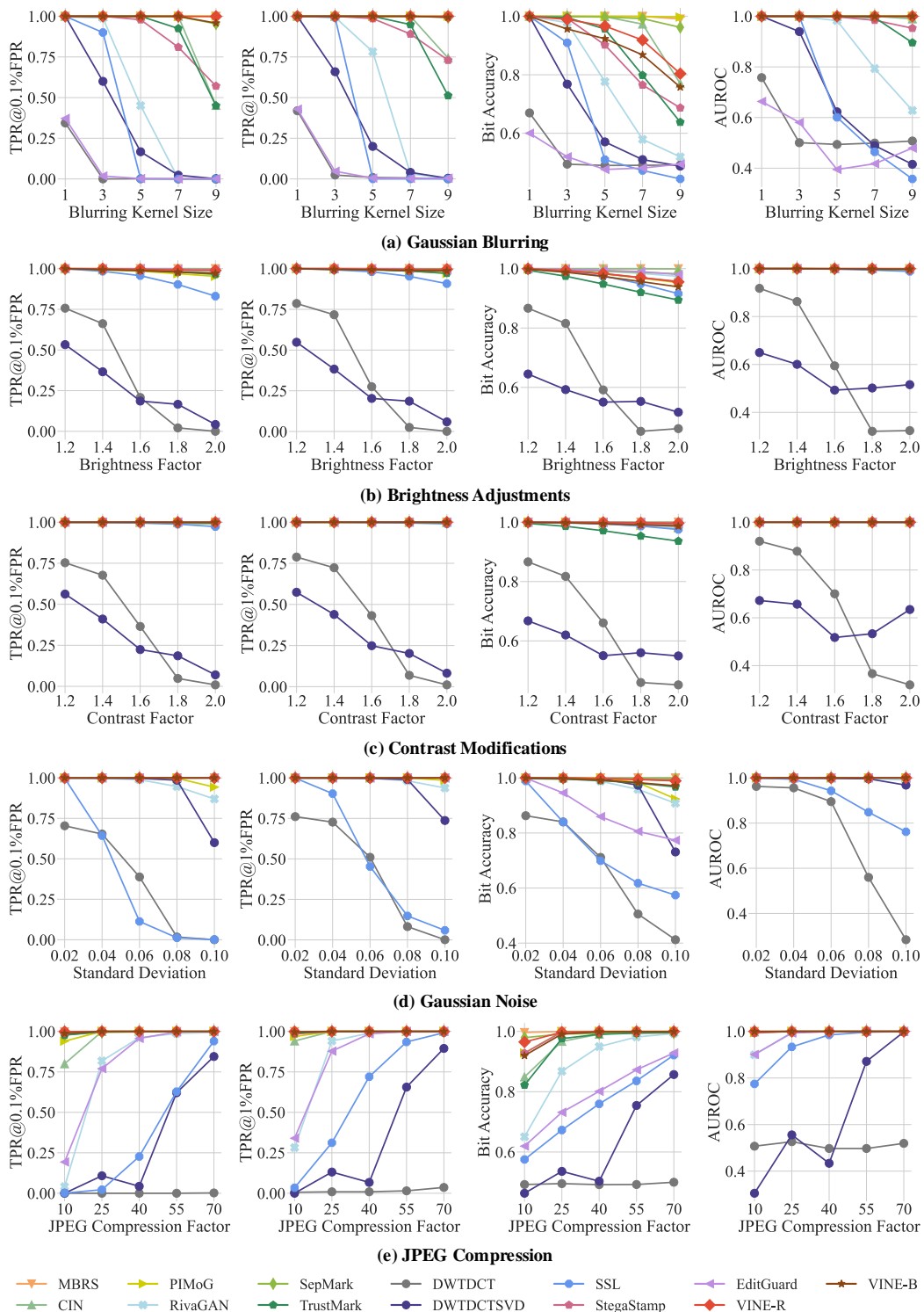

Figure 9: Performance of watermarking methods at a resolution of 512×512 pixels under (a) Gaussian blurring, (b) brightness adjustments, (c) contrast modifications, (d) Gaussian noise, and (e) JPEG compression.

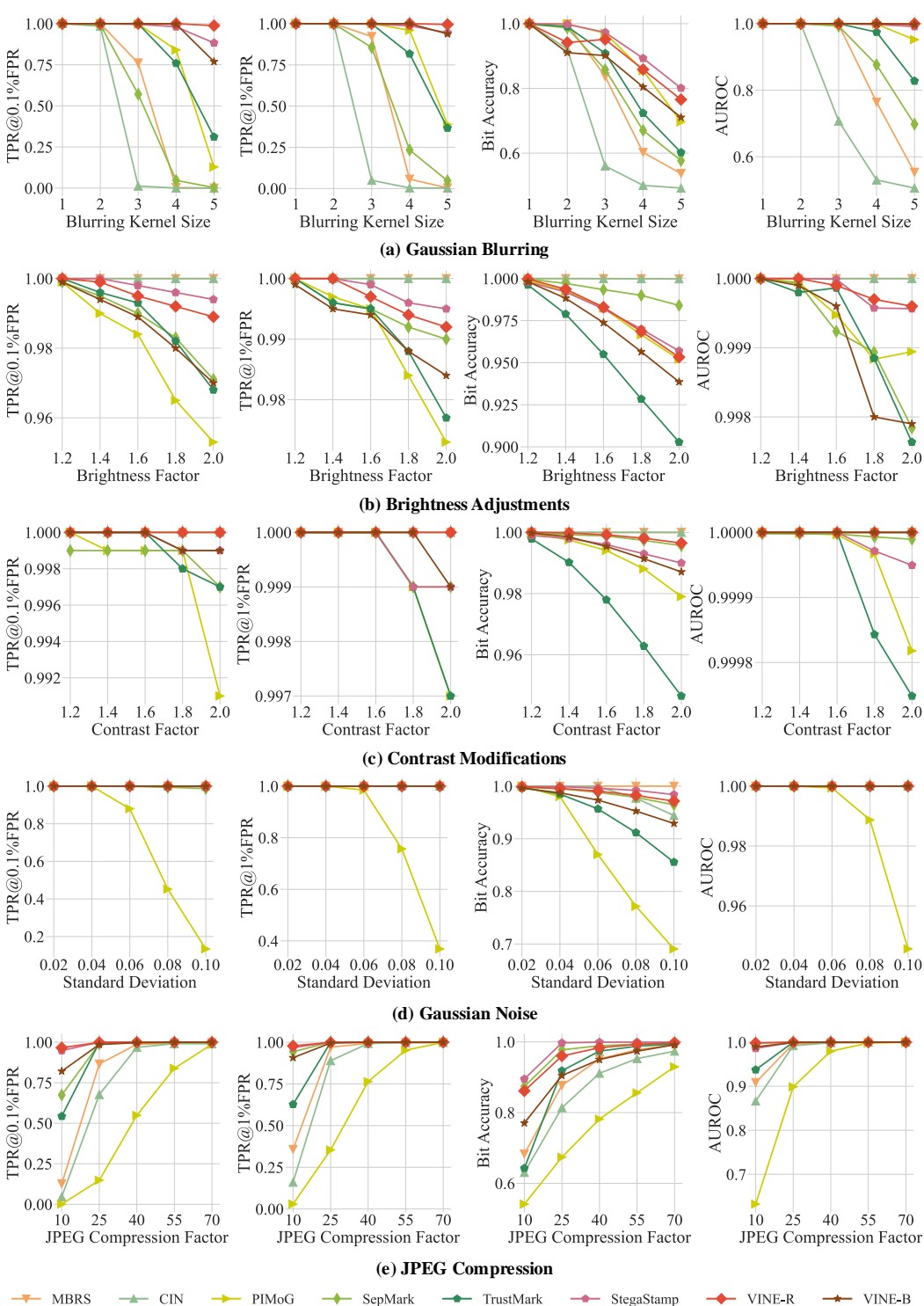

Figure 10: Assessment of watermarking methods at their respective training resolutions under the following conditions: (a) Gaussian blurring, (b) brightness adjustments, (c) contrast modifications, (d) Gaussian noise, and (e) JPEG compression. Training resolutions: MBRS, CIN, PIMoG, and SepMark were trained at 128×128 pixels; TrustMark, VINE-B, and VINE-R at 256×256 pixels; and StegaStamp at 400×400 pixels.

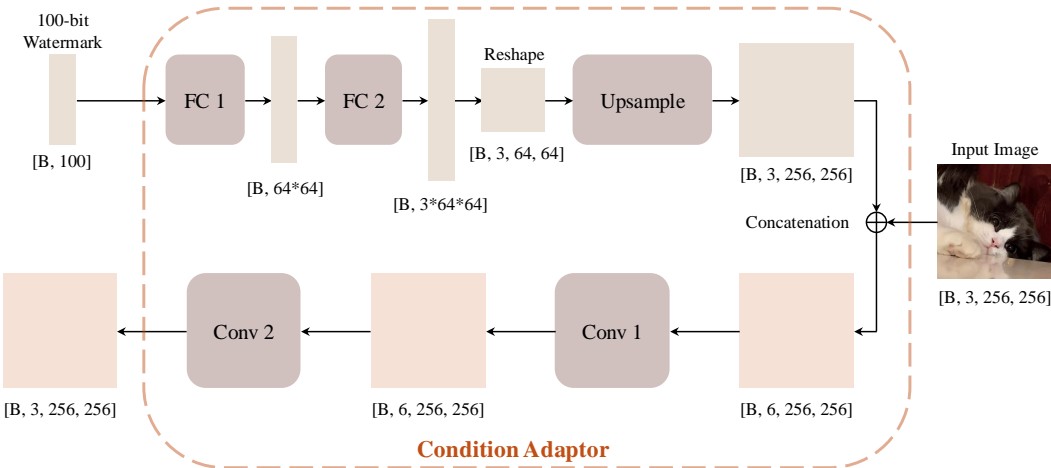

Figure 11: Architecture of the condition adaptor in Figure 4. Each fully connected and convolutional layer is followed by an activation layer.

# G    ANALYSIS OF IMAGE EDITING

## G.1    RATIONALE FOR IMAGE EDITING SELECTION

Image regeneration, global editing, local editing, and image-to-video generation cover the majority of editing needs. Image editing can be broadly divided into global and local categories, as user edits typically affect either the entire image or specific parts of it.

Global editing involves altering most of an image's pixels while maintaining its overall layout, components, or semantics. Techniques such as style transfer, cartoonization, image translation, and scene transformation fall under this category and produce similar effects. For example, using prompts like 'turn it into Van Gogh style' or 'convert it into a sketchy painting' in an image editing model can effectively achieve style transfer.

Local editing, on the other hand, refers to modifications applied to specific elements, semantics, or regions within an image. This category includes image inpainting, image composition, object manipulation, attribute manipulation, and so forth.

While image regeneration and image-to-video generation are not strictly considered forms of image editing, they can be used to create similar digital content while removing watermarks, thereby posing a threat to copyright protection. For this reason, we have included them in our benchmark.

## G.2    IMAGE REGENERATION

In this study, we assess the reconstruction quality of image regeneration across various difficulty levels. As illustrated in Figure 12, increasing the difficulty level—by either raising the noise timestep in stochastic regeneration or decreasing the sampling step in deterministic regeneration—degrades the reconstruction quality. Figure 13 presents several examples. The regenerated images are perceptually similar to the original images. However, as the difficulty level increases, stochastic regeneration tends to introduce more hallucinated details into the original image, whereas deterministic regeneration tends to smooth out the original images. Both approaches can degrade the embedded watermark, thereby posing challenges to copyright protection. Our method effectively embeds and detects watermarks even under extremely challenging difficulty levels.

## G.3    GLOBAL EDITING AND LOCAL EDITING

Here, we assess the editing quality of various global editing models using three metrics: $\text{CLIP}_{\text{dir}}$, $\text{CLIP}_{\text{out}}$, and $\text{CLIP}_{\text{img}}$. $\text{CLIP}_{\text{dir}}$ evaluates whether the edits correspond to the modifications specified by the editing prompt, which represents the difference between the captions of the source and target

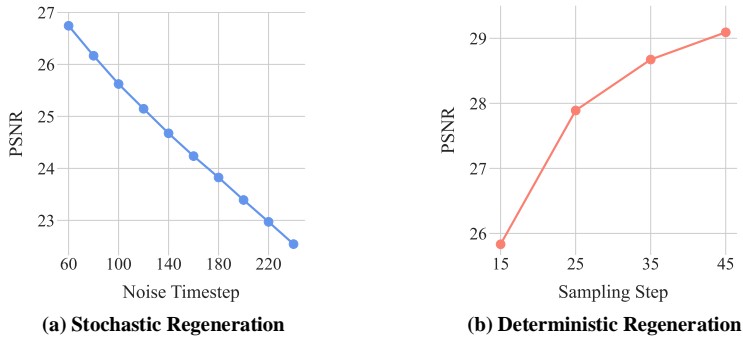

Figure 12: The reconstruction quality of (a) stochastic regeneration and (b) deterministic regeneration. The PSNR is calculated by comparing the regenerated image to the original image.

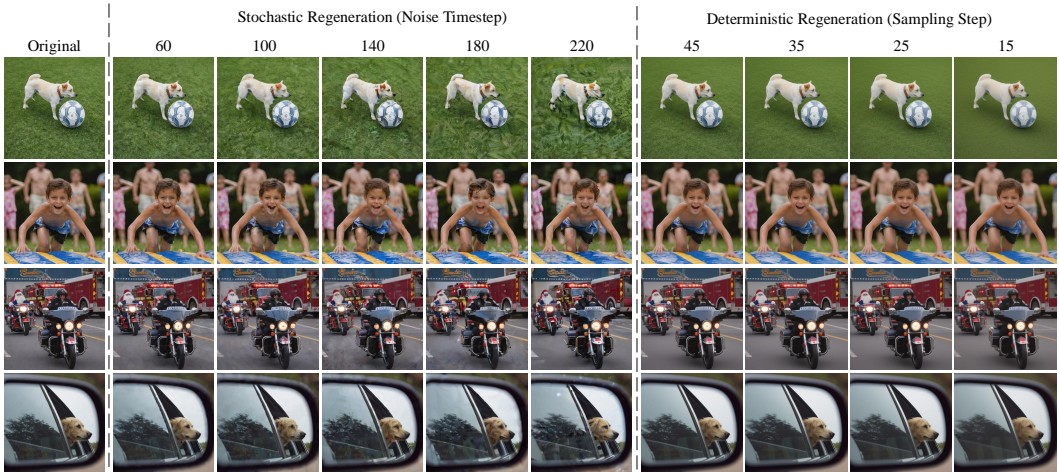

Figure 13: The reconstruction quality of stochastic regeneration and deterministic regeneration. Please zoom in for a closer look.

images. This metric is calculated by measuring the cosine similarity between the difference in CLIP image embeddings of the source and edited images and the difference in CLIP text embeddings of the source and target captions. $\text{CLIP}_{\text{out}}$ measures the alignment of the edited image with the target caption by calculating the cosine similarity between the CLIP text embeddings of the target caption and the CLIP image embeddings of the edited images. Meanwhile, $\text{CLIP}_{\text{img}}$ assesses content preservation by evaluating the cosine similarity between the CLIP image embeddings of the source and edited images. Table 4 presents a comparison of editing results from different editing models applied to unwatermarked images in the first row. Each editing model employs an image guidance scale of 1.5 and a text guidance scale of 7. Within these settings, UltraEdit most effectively aligns the edited image with the editing prompt, while MagicBrush excels at preserving the image layout.

Since text-driven image editing is more sensitive than image regeneration, we investigated whether watermarking the input image affects editing quality. The results, presented in Table 4, show that editing quality remains largely intact with only minor fluctuations. If the watermark degrades the image quality, it can slightly impact editing performance in terms of both text alignment and content preservation. Figure 14 shows several examples. Similarly, the evaluation results for local editing models are presented in Table 5 and Figure 15. UltraEdit also demonstrates superior editing quality for local edits, excelling in both text-image alignment and image content preservation.

Table 4: Comparison of editing quality for different global editing methods and the effect of different watermarks on image editing outcomes. All models use an image guidance scale of 1.5 and a text guidance scale of 7.

| Method | Instruct-Pix2Pix | | | UltraEdit | | | MagicBrush | | |
|---|---|---|---|---|---|---|---|---|---|
| | $\text{CLIP}_{dir}$ ↑ | $\text{CLIP}_{img}$ ↑ | $\text{CLIP}_{out}$ ↑ | $\text{CLIP}_{dir}$ ↑ | $\text{CLIP}_{img}$ ↑ | $\text{CLIP}_{out}$ ↑ | $\text{CLIP}_{dir}$ ↑ | $\text{CLIP}_{img}$ ↑ | $\text{CLIP}_{out}$ ↑ |
| Unwatermarked Image | 0.2693 | 0.7283 | 0.2732 | 0.3230 | 0.7268 | 0.3008 | 0.3025 | 0.7913 | 0.2930 |
| MBRS (Jia et al., 2021) | 0.2494 | 0.7385 | 0.2733 | 0.2919 | 0.6654 | 0.2891 | 0.2857 | 0.7816 | 0.2929 |
| CIN (Ma et al., 2022) | 0.2625 | 0.7232 | 0.2729 | 0.3152 | 0.7111 | 0.3010 | 0.2949 | 0.7841 | 0.2928 |
| PIMoG (Fang et al., 2022) | 0.2518 | 0.7021 | 0.2746 | 0.3010 | 0.6940 | 0.3024 | 0.2815 | 0.7662 | 0.2962 |
| RivaGAN (Zhang et al., 2019) | 0.2647 | 0.7317 | 0.2721 | 0.3168 | 0.7133 | 0.3003 | 0.3020 | 0.7948 | 0.2930 |
| SepMark (Wu et al., 2023) | 0.2659 | 0.7292 | 0.2743 | 0.3145 | 0.7181 | 0.3002 | 0.2975 | 0.7891 | 0.2936 |
| DWTDCT (Al-Haj, 2007) | 0.2644 | 0.7317 | 0.2734 | 0.3189 | 0.7250 | 0.3009 | 0.2959 | 0.7942 | 0.2934 |
| DWTDCTSVD (Navas et al., 2008) | 0.2581 | 0.7220 | 0.2751 | 0.3115 | 0.7118 | 0.3004 | 0.2869 | 0.7793 | 0.2939 |
| SSL (Fernandez et al., 2022) | 0.2583 | 0.7218 | 0.2752 | 0.3093 | 0.7065 | 0.3019 | 0.2896 | 0.7780 | 0.2944 |
| StegaStamp (Tancik et al., 2020) | 0.2436 | 0.6826 | 0.2697 | 0.2904 | 0.6886 | 0.3007 | 0.2663 | 0.7512 | 0.2944 |
| TrustMark (Bui et al., 2023) | 0.2634 | 0.7181 | 0.2729 | 0.3172 | 0.7146 | 0.2994 | 0.2943 | 0.7853 | 0.2936 |
| EditGuard (Zhang et al., 2024d) | 0.2722 | 0.7045 | 0.2722 | 0.3155 | 0.7170 | 0.3021 | 0.2882 | 0.7708 | 0.2940 |
| VINE-Base | 0.2743 | 0.7260 | 0.2743 | 0.3186 | 0.7189 | 0.2996 | 0.2977 | 0.7889 | 0.2931 |
| VINE-Robust | 0.2624 | 0.7248 | 0.2715 | 0.3176 | 0.7183 | 0.3001 | 0.2981 | 0.7953 | 0.2940 |

Table 5: Comparison of editing quality for different local editing methods and the effect of different watermarks on image editing outcomes. All models use an image guidance scale of 1.5 and a text guidance scale of 7.

| Method | ControlNet-Inpainting | | | UltraEdit | | |
|---|---|---|---|---|---|---|
| | $\text{CLIP}_{dir}$ ↑ | $\text{CLIP}_{img}$ ↑ | $\text{CLIP}_{out}$ ↑ | $\text{CLIP}_{dir}$ ↑ | $\text{CLIP}_{img}$ ↑ | $\text{CLIP}_{out}$ ↑ |
| Unwatermarked Image | 0.1983 | 0.7076 | 0.2589 | 0.2778 | 0.7519 | 0.2917 |
| MBRS (Jia et al., 2021) | 0.1846 | 0.7058 | 0.2588 | 0.2657 | 0.7175 | 0.2913 |
| CIN (Ma et al., 2022) | 0.1966 | 0.7042 | 0.2613 | 0.2745 | 0.7389 | 0.2922 |
| PIMoG (Fang et al., 2022) | 0.1828 | 0.6909 | 0.2600 | 0.2578 | 0.7371 | 0.2920 |
| RivaGAN (Zhang et al., 2019) | 0.1975 | 0.7117 | 0.2612 | 0.2748 | 0.7469 | 0.2937 |
| SepMark (Wu et al., 2023) | 0.1932 | 0.7126 | 0.2582 | 0.2716 | 0.7588 | 0.2921 |
| DWTDCT (Al-Haj, 2007) | 0.1982 | 0.7197 | 0.2602 | 0.2776 | 0.7558 | 0.2924 |
| DWTDCTSVD (Navas et al., 2008) | 0.1922 | 0.6995 | 0.2608 | 0.2705 | 0.7469 | 0.2940 |
| SSL (Fernandez et al., 2022) | 0.1911 | 0.6995 | 0.2604 | 0.2677 | 0.7380 | 0.2940 |
| StegaStamp (Tancik et al., 2020) | 0.1752 | 0.6684 | 0.2606 | 0.2439 | 0.7246 | 0.2919 |
| TrustMark (Bui et al., 2023) | 0.1959 | 0.7001 | 0.2594 | 0.2728 | 0.7451 | 0.2919 |
| EditGuard (Zhang et al., 2024d) | 0.1921 | 0.6944 | 0.2606 | 0.2696 | 0.7392 | 0.2923 |
| VINE-Base | 0.1953 | 0.7023 | 0.2591 | 0.2726 | 0.7494 | 0.2906 |
| VINE-Robust | 0.1951 | 0.7030 | 0.2591 | 0.2710 | 0.7475 | 0.2909 |

Table 6: Comparison of watermarking performance in terms of detection accuracy on image-to-video generation with MAGE+. TPR@0.1%FPR is averaged over 1,000 videos.

| Method | TPR@0.1%FPR ↑ (%) | | | | | |
|---|---|---|---|---|---|---|
| | Frame 2 | Frame 4 | Frame 6 | Frame 8 | Frame 10 | Average |
| MBRS (Jia et al., 2021) | 89.57 | 88.67 | 87.45 | 86.51 | 84.42 | 87.32 |
| CIN (Ma et al., 2022) | 45.92 | 44.78 | 43.21 | 42.17 | 40.71 | 43.36 |
| PIMoG (Fang et al., 2022) | 78.23 | 76.99 | 75.72 | 74.91 | 73.12 | 75.79 |
| RivaGAN (Zhang et al., 2019) | 56.87 | 54.83 | 53.21 | 52.14 | 51.01 | 53.61 |
| SepMark (Wu et al., 2023) | 63.45 | 62.15 | 61.03 | 60.89 | 59.24 | 61.35 |
| DWTDCT (Al-Haj, 2007) | 30.57 | 29.51 | 28.89 | 27.72 | 26.87 | 28.71 |
| DWTDCTSVD (Navas et al., 2008) | 38.56 | 38.54 | 37.12 | 36.81 | 35.74 | 37.35 |
| SSL (Fernandez et al., 2022) | 81.21 | 80.95 | 78.99 | 77.18 | 76.12 | 78.89 |
| StegaStamp (Tancik et al., 2020) | 91.25 | 90.34 | 89.12 | 88.67 | 87.33 | 89.34 |
| TrustMark (Bui et al., 2023) | 90.35 | 90.12 | 89.45 | 87.69 | 86.13 | 88.75 |
| EditGuard (Zhang et al., 2024d) | 42.57 | 41.46 | 40.55 | 39.91 | 38.17 | 40.53 |
| VINE-Base | 92.22 | 91.35 | 90.74 | 89.12 | 88.01 | 90.29 |
| VINE-Robust | 93.14 | 92.88 | 91.32 | 90.27 | 89.12 | 91.35 |

### G.4 IMAGE-TO-VIDEO GENERATION

Figure 16 presents the results of image-to-video generation applied to watermarked images using various watermarking methods. The presence of watermarks does not perceptually affect the image-to-video generation.

Additionally, we add watermarks to 1,000 images from the CATER-GEN-v2 dataset Hu et al. (2022) and use the MAGE+ model Hu et al. (2022) to perform text-image-to-video (TI2V) generation, producing 10-frame videos. For watermark detection, we analyze frames 2, 4, 6, 8, and 10. The average detection accuracies across 1,000 videos are presented in Table 6.

## H INFERENCE SPEED AND GPU MEMORY EVALUATION

We evaluate the inference speed and GPU memory usage of watermarking methods on an NVIDIA Quadro RTX6000 GPU. The results are averaged over 1,000 images. SSL employs instance-based iterative optimization, resulting in longer processing times. EditGuard utilizes an inverse neural network to encode secret images and bits simultaneously, thereby also requiring additional running time. Although our model demands more GPU memory and longer inference times, these requirements remain within acceptable ranges.

Table 7: Comparison of watermarking methods based on running time per single image and GPU memory usage. The results are averaged over 1,000 images. Since the implementations we employed for DWTDCT, DWTDCTSVD, and RivaGAN support CPUs exclusively, they have been omitted from the comparison.

| Method | Running Time per Image (s) | GPU Memory Usage (MB) |
|---|---|---|
| MBRS (Jia et al., 2021) | 0.0053 | 938 |
| CIN (Ma et al., 2022) | 0.0741 | 2944 |
| PIMoG (Fang et al., 2022) | 0.0212 | 878 |
| RivaGAN (Zhang et al., 2019) | - | - |
| SepMark (Wu et al., 2023) | 0.0109 | 928 |
| DWTDCT (Al-Haj, 2007) | - | - |
| DWTDCTSVD (Navas et al., 2008) | - | - |
| SSL (Fernandez et al., 2022) | 2.1938 | 1072 |
| StegaStamp (Tancik et al., 2020) | 0.0672 | 1984 |
| TrustMark (Bui et al., 2023) | 0.0705 | 648 |
| EditGuard (Zhang et al., 2024d) | 0.2423 | 1638 |
| VINE | 0.0795 | 4982 |

## I QUALITATIVE COMPARISON

Figure 17 showcases qualitative examples of $512 \times 512$ images encoded using the watermarking methods evaluated in this study. The residuals are calculated by normalizing the absolute difference $|\boldsymbol{x}_w - \boldsymbol{x}_o|$. For a more detailed examination, please zoom in on the images. It can be observed that the images watermarked by MBRS and PIMoG exhibit slight color distortions, whereas the StegaStamp watermarked images display black artifacts. Other methods produce watermarked images that, to the human eye, appear identical to the original images.

## J ADDITIONAL BENCHMARKING RESULTS

Additional benchmarking results are presented in Figure 18, including deterministic regeneration with DPM-Solver, global editing with UltraEdit, global editing with Instruct-Pix2Pix, local editing with ControlNet-Inpainting, and image-to-video generation with Stable Video Diffusion.

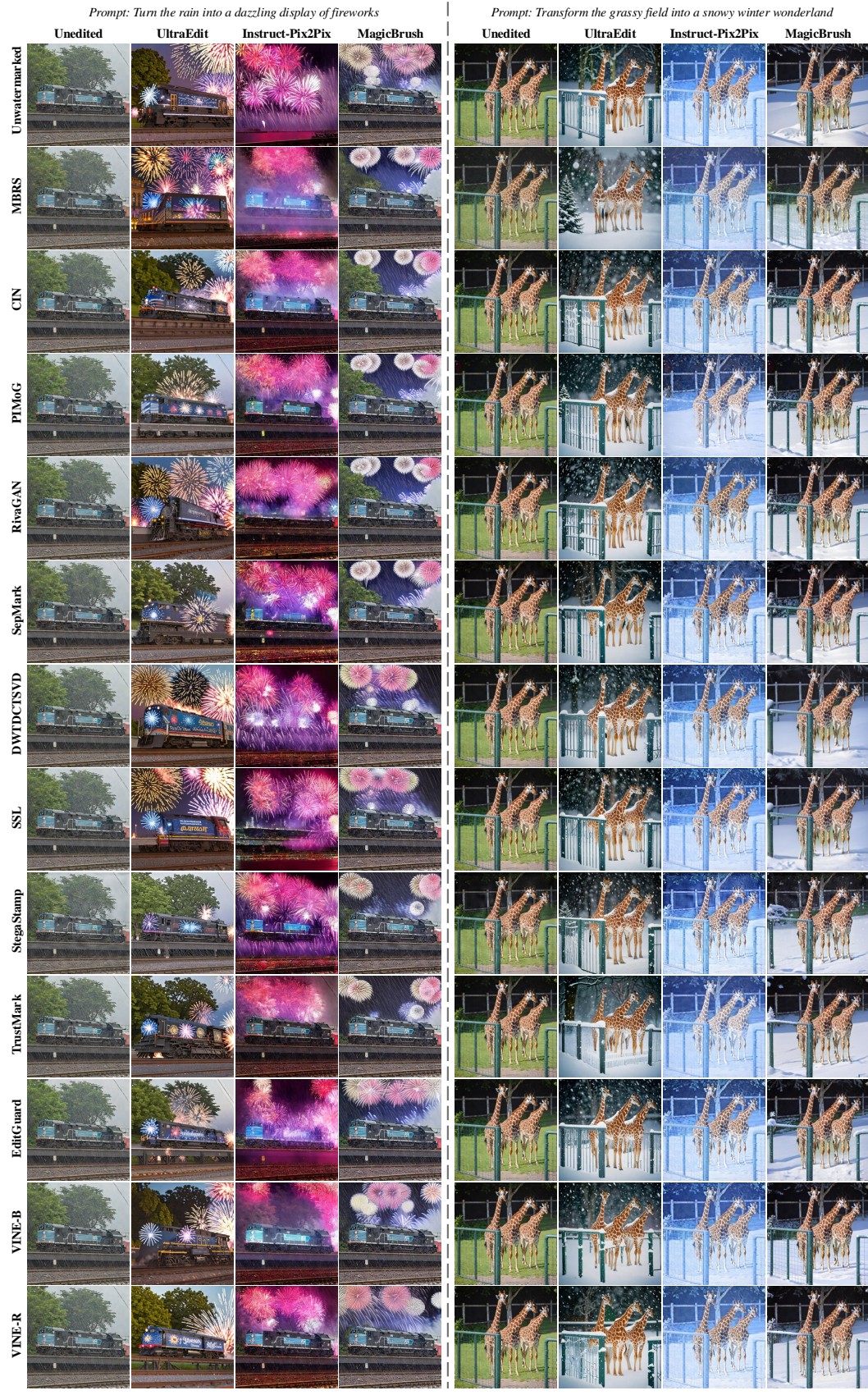

Figure 14: Different watermarks have minimal impact on the image global editing outcomes, resulting in only slight changes.

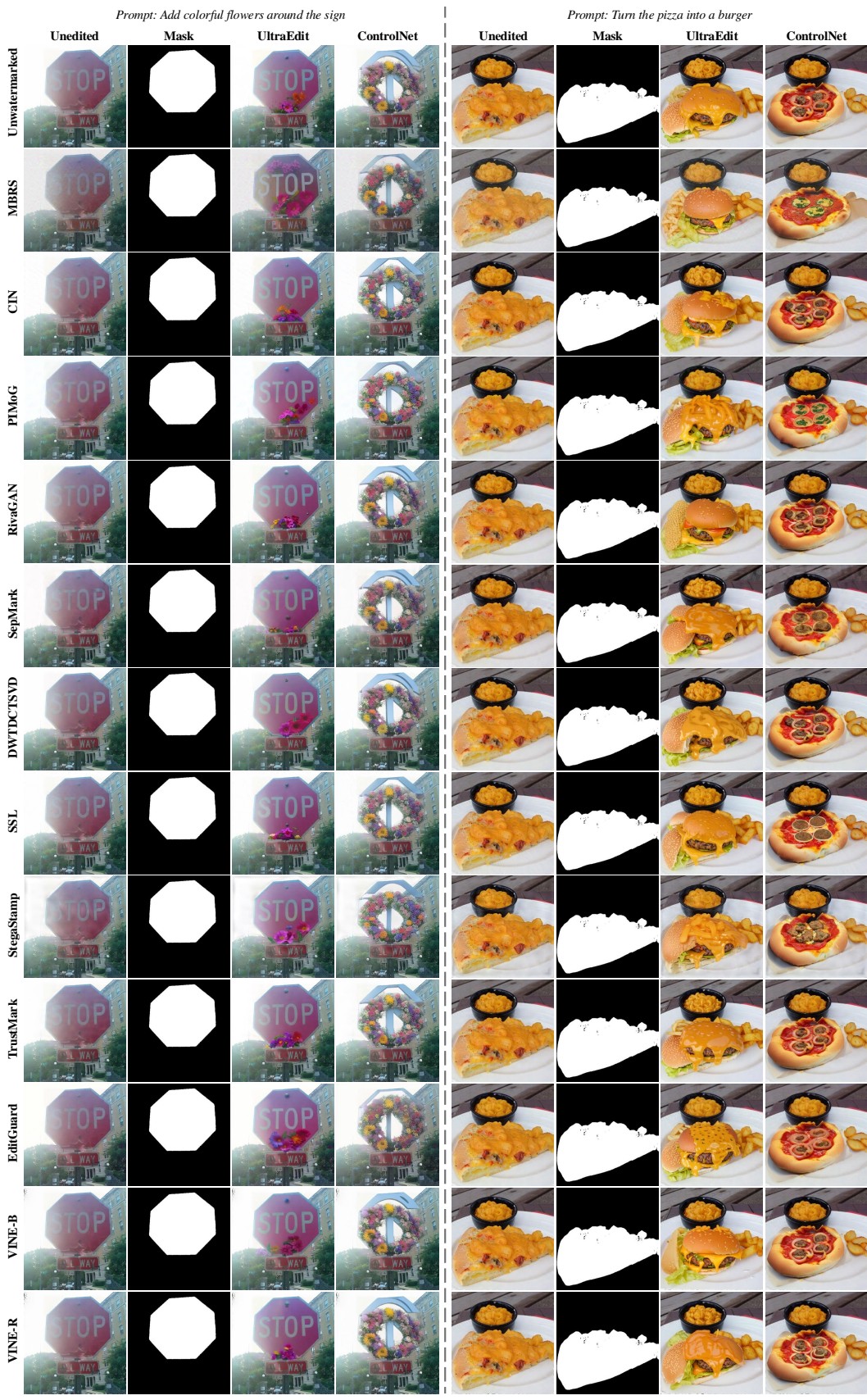

Figure 15: Different watermarks have minimal impact on the image local editing outcomes, resulting in only slight changes.

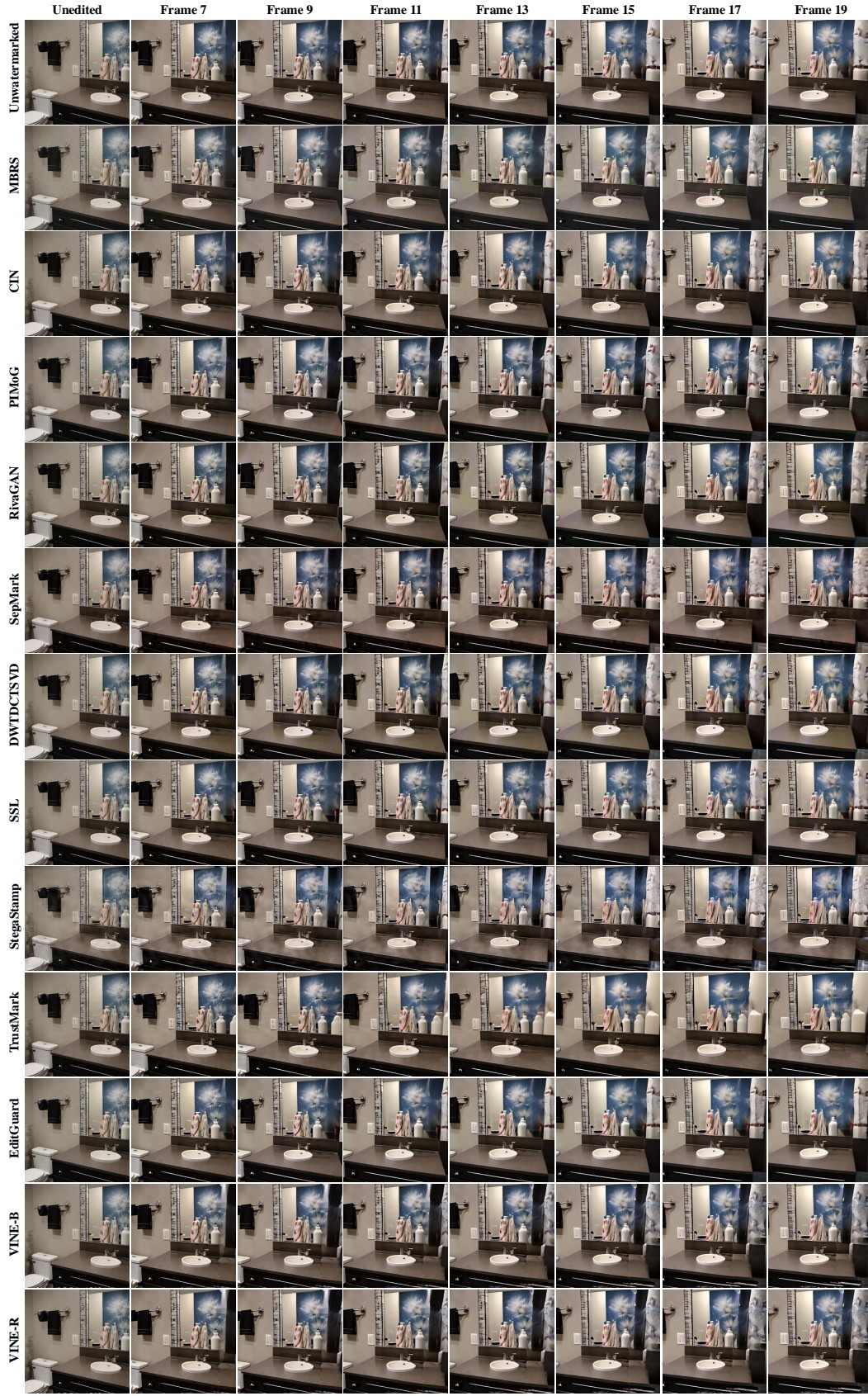

Figure 16: Different watermarks have little effect on image-to-video generation, leading to only minor changes.

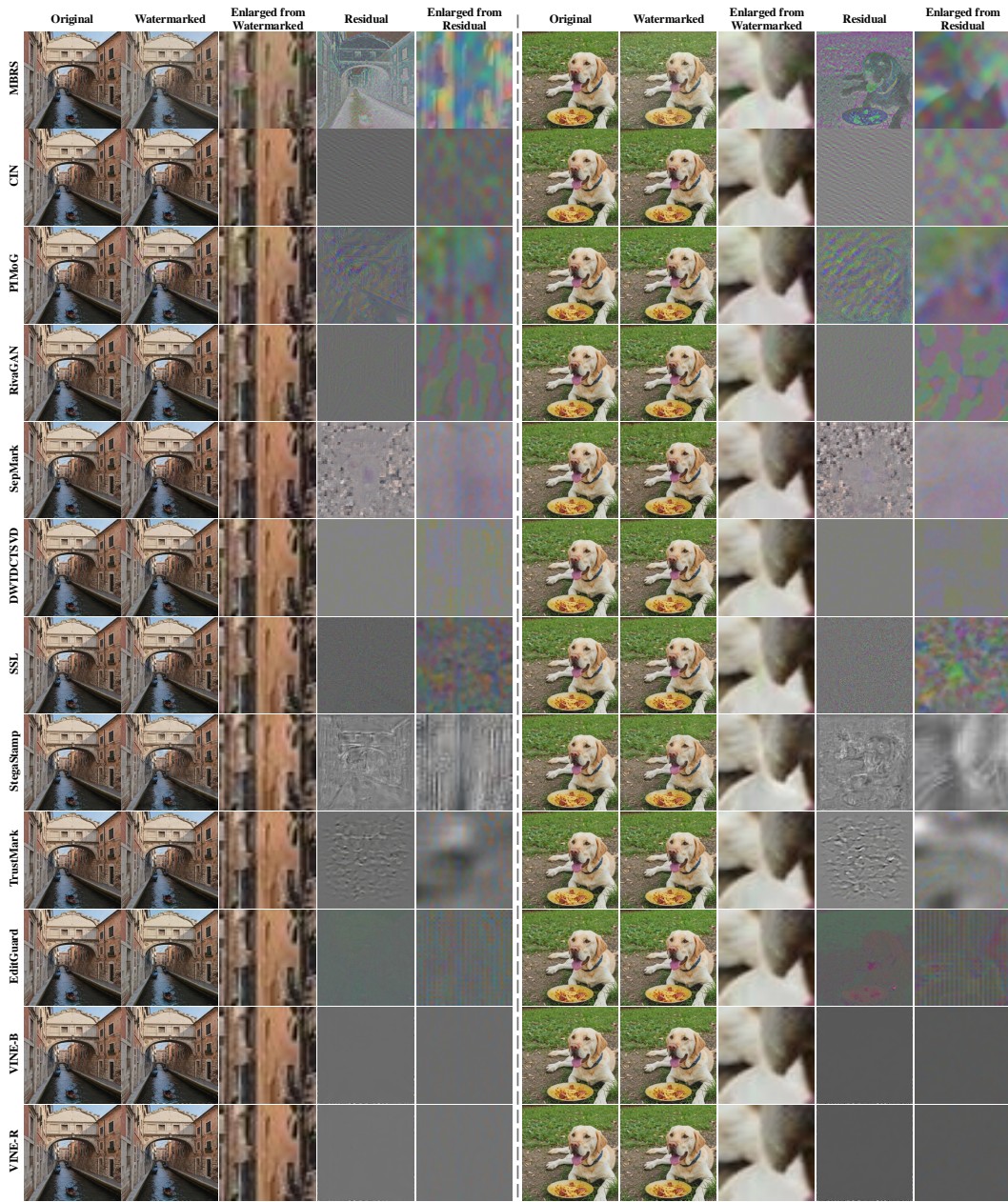

Figure 17: Qualitative comparison of the evaluated watermarking methods. The 'Enlarged from Watermark' column displays a magnified $40 \times 40$ central region of the watermarked images, while the 'Enlarged from Residual' column shows a magnified $40 \times 40$ central region of the residual images. Please zoom in for a closer examination.

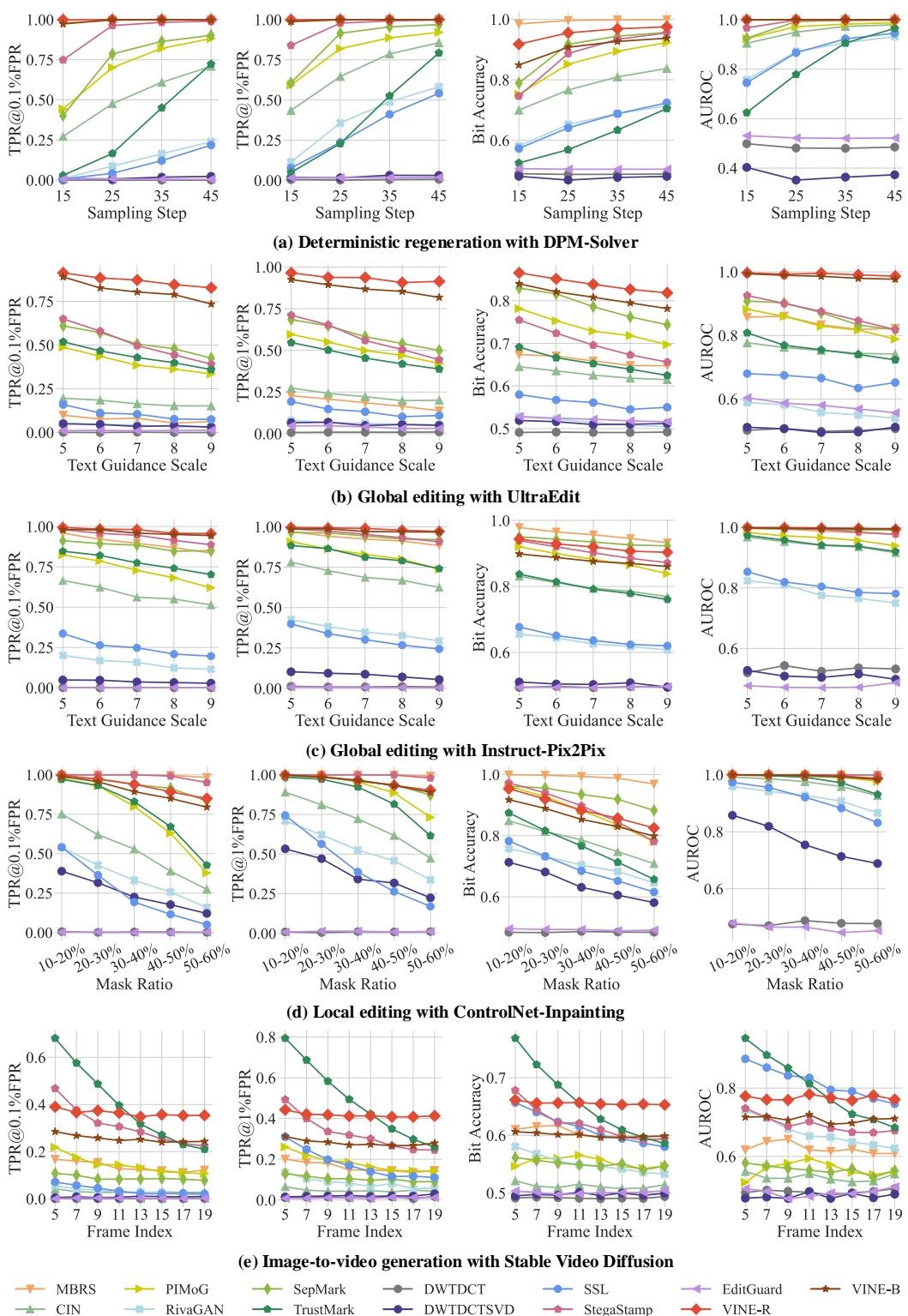

Figure 18: The performance of watermarking methods under (a) Deterministic regeneration with DPM-Solver, (b) Global editing with UltraEdit, (c) Global editing with Instruct-Pix2Pix, (d) Local editing with ControlNet-Inpainting, and (e) Image-to-video generation with Stable Video Diffusion.

