# OpenReview forum: "Robust Watermarking Using Generative Priors Against Image Editing: From Benchmarking to Advances"
_ICLR.cc/2025/Conference — ICLR 2025 Poster_

### Official Review · Reviewer_CgKQ · 2024-11-03

**Soundness:** 2
**Presentation:** 3
**Contribution:** 2
**Rating:** 6
**Confidence:** 4

**Summary:**

The paper evaluates eleven watermarking methods against prevalent image editing techniques and demonstrates that most methods fail to detect watermarks after such edits. It also introduces a watermarking model based on SDXL-Turbo, which exhibits high robustness against these editing methods while maintaining high image quality.

**Strengths:**

The paper presents the first holistic benchmark that incorporates four types of image editing techniques to assess the robustness of watermarking methods. This is significant for evaluating the robustness of future watermarking methods, as it helps to promote the standardization and comprehensiveness of robustness assessments. By addressing a critical gap in evaluating watermark resilience against sophisticated transformations enabled by modern generative models, this work encourages researchers in the field of image watermarking to focus on the robustness of their methods against emerging image editing technologies, including image regeneration, global editing, local editing, and image-to-video generation. Overall, the paper is clearly articulated and well-supported.

**Weaknesses:**

1. The paper explains the reasons behind the watermarking algorithm's resistance to image editing from the perspective of the frequency domain. It notes that the watermarking methods exhibiting high robustness against image editing in certain scenarios display prominent patterns in the low-frequency bands, which aligns with the general understanding of watermark robustness. However, the paper primarily focuses on the robustness of watermarking methods against image editing techniques based on generative models. Therefore, summarizing the unique effects of such image editing techniques on the watermark is more meaningful.
2. We observe that the proposed watermarking method, VINE, shows higher brightness in the central region of the frequency domain, which corresponds to the author's analysis of watermark robustness. However, the paper does not clarify why this watermarking model based on SDXL-Turbo exhibits such characteristics, leading to the author's specific design of the watermark algorithm. In other words, there seems to be a disconnect between the author's analysis of watermark robustness and the design of the watermark model.

**Questions:**

1.Figure 6 in the appendix shows that VINE exhibits higher brightness in the central region, providing evidence for why the proposed watermarking method demonstrates strong robustness against image editing. If the author can thoroughly elucidate the principles underlying this phenomenon, it may address the previously mentioned issue of "a disconnect between the author's analysis of watermark robustness and the design of the watermark model."

2.The experimental results demonstrate that the proposed watermarking method, VINE, significantly enhances robustness against various image editing techniques. Has the author considered using representative image editing as an attack template, incorporating the associated attack loss as one of the objective functions during the training phase? Alternatively, how might integrating the specific effects of image editing on watermarks into the design of the watermarking model influence the results of the watermarking algorithm?

3. In the experimental section, some of the differences between the subjective experimental results are difficult to discern visually. The author could consider selecting a subset of images and enlarging specific regions to facilitate reader comprehension.

---

> ### Author Response · Authors · 2024-11-21
> **Response to Reviewer CgKQ (1/2)**
>
> We sincerely appreciate your recognition of the significance of our benchmark and its benefits for future research, as well as your acknowledgment that our paper is clearly articulated and well-supported!
>
> We will address all the concerns point by point.
>
> ***
>
> **Weakness 1: The paper explains the reasons behind the watermarking algorithm's resistance to image editing from the perspective of the frequency domain. It notes that the watermarking methods exhibiting high robustness against image editing in certain scenarios display prominent patterns in the low-frequency bands, which aligns with the general understanding of watermark robustness. However, the paper primarily focuses on the robustness of watermarking methods against image editing techniques based on generative models. Therefore, summarizing the unique effects of such image editing techniques on the watermark is more meaningful.**
>
> Thank you for your valuable suggestions.
>
> In response to your feedback, we added a new Figure 8 in Appendix B (in the new version of our submission) to illustrate the impact of different image editing methods on various watermark patterns within the frequency domain. As shown, the frequency patterns of VINE-R, VINE-B, MBRS, and StegaStamp are less affected compared to their original patterns (shown in Figure 7) than those of other watermarking methods. We believe this provides interesting and insightful information for our readers, and we are truly grateful for your input in enhancing the clarity of our work.
>
> Additionally, we retain Figure 3 to help readers better understand the impact of image editing in a more disentangled manner, as it can be challenging to isolate the effects on low-, mid-, and high-frequency bands solely through the visual representation in Figure 8.
>
> If any aspects remain unclear or if further clarification of other figures is needed, we would be happy to make additional improvements!
>
> ***
> **Weakness 2 & Question 1: Figure 6 in the appendix shows that VINE exhibits higher brightness in the central region, providing evidence for why the proposed watermarking method demonstrates strong robustness against image editing. If the author can thoroughly elucidate the principles underlying this phenomenon, it may address the previously mentioned issue of "a disconnect between the author's analysis of watermark robustness and the design of the watermark model."**
>
> Thank you for your insightful question. The observation that VINE exhibits higher brightness in the low-frequency bands is indeed interesting, though this was not the motivation behind our design choices. Allow us to walk you through our design and analysis process.
>
> Our two key design elements (surrogate layer & generative prior adaptation) are based on the two following observations: (1) Image editing and image blurring exhibit similar frequency characteristics. (2) Image watermarking can be viewed as a form of conditional generation, where a generative prior can enhance image quality by making watermarks less visible. The surrogate layer enhances our model's robustness to image editing, while the generative prior improves the quality of the encoded images. Therefore, our design is grounded in these analyses.
>
> Regarding the intriguing correlation you mentioned—the robustness of our watermarking model against image editing being highly positively correlated with pattern intensity in the low-frequency region—this emerged from our training process rather than driving our design decisions.
>
> In summary, this finding is a byproduct of our design rather than a deliberate motivation of it. We did not intend to concentrate the watermark pattern in the low-frequency region. In fact, when we attempted to do so by introducing an additional frequency loss, it resulted in poorer performance, which we discuss in the next question. This finding could offer valuable insights for future research aimed at achieving a better balance between robustness and encoded image quality.
>
> We have included this discussion in Appendix B of the new version of our submission (highlighted in green) to enhance readers' understanding. Thank you for helping to make our work more complete!

---

> ### Author Response · Authors · 2024-11-21
> **Response to Reviewer CgKQ (2/2)**
>
> **Question 2: (1) The experimental results demonstrate that the proposed watermarking method, VINE, significantly enhances robustness against various image editing techniques. Has the author considered using representative image editing as an attack template, incorporating the associated attack loss as one of the objective functions during the training phase? (2) Alternatively, how might integrating the specific effects of image editing on watermarks into the design of the watermarking model influence the results of the watermarking algorithm?**
>
> (1) We have considered incorporating a representative image editing method into our noise layer during training. This approach has led to the development of our VINE-R variant, which is fine-tuned from VINE-B by integrating the Instruct-Pix2Pix pipeline during the training process.
>
> (2) Image editing tends to remove patterns present in high-frequency bands. Thus, in our preliminary design, we expected that integrating the effects of image editing into our design would involve deliberately forcing the pattern to appear in the low-frequency band by adding an additional loss in the frequency domain. This point is particularly intriguing.
>
> In our preliminary experiments, we attempted to position the pattern in the low-frequency region by ensuring that the high-frequency bands of the Fourier spectra of the watermarked and original images are identical. However, this loss did not lead to a significant improvement in robustness and resulted in a substantial decrease in the quality of the encoded images.
>
> We believe this outcome occurs because injecting all information into the low-frequency bands adversely affects the image quality. In contrast, allowing the model the flexibility to choose which frequency bands to inject the watermark helps balance robustness and image quality. Therefore, we decided not to include this frequency loss in our final design.
>
> ***
> **Question 3: In the experimental section, some of the differences between the subjective experimental results are difficult to discern visually. The author could consider selecting a subset of images and enlarging specific regions to facilitate reader comprehension.**
>
> Thank you for your valuable suggestion! Based on your reference to the "subjective experimental results," we believe you are referring to Figure 15 in the first version of our submission. In the revised submission, this figure has been renumbered to Figure 17 and enhanced with two additional columns that display enlarged central $40\times40$ regions of the watermarked and residual images, respectively. If anything else is unclear or if other figures require further clarification, we are happy to make additional improvements.
>
> ***
> We hope this addresses the concerns you may have and are always available for further discussion. We deeply appreciate the time you have taken to engage with our work and share your valuable insights.

---

> > ### Author Response · Authors · 2024-11-26
> >
> > Dear Reviewer CgKQ,
> >
> > We sincerely appreciate the time and effort you have dedicated to reviewing our paper.
> >
> > As the discussion period approaches its end, we would like to gently remind you that there are seven days remaining for any additional comments or questions. We would be grateful for the opportunity to address any further concerns you may have before the discussion phase concludes.
> >
> > Thank you very much!
> >
> > Many thanks,
> >
> > The Authors

---

> > ### Comment · Reviewer_CgKQ · 2024-11-27
> >
> > Thank you for your detailed response. It has effectively addressed my concerns, allowing me to better appreciate and accept your viewpoint.  I would like to increase the score.

---

> > > ### Author Response · Authors · 2024-11-27
> > >
> > > Thank you for your positive feedback on our revisions. We also deeply appreciate your valuable contributions throughout the review process!

---

### Official Review · Reviewer_biAS · 2024-11-03

**Soundness:** 3
**Presentation:** 3
**Contribution:** 3
**Rating:** 8
**Confidence:** 3

**Summary:**

This paper introduces an image watermarking benchmark, specifically aiming to evaluate the watermark robustness against four image editing methods. In addition, an image watermarking that is robust against image editing is proposed.

**Strengths:**

1. This paper focuses on the image watermark robustness against image editing, which is important but has rarely been explored.
2. The proposed benchmark includes different types of image editing approaches, rendering it comprehensive to some extent.
3. The proposed SDXL-Turbo-based robust image watermarking method is novel, and the experiments demonstrate its effectiveness.
4. The paper is overall well-written.

**Weaknesses:**

1. The benchmark only considers four types of image editing methods (image regeneration, global editing, local editing, and image-to-video generation). Other image editing methods such as style transfer are not considered.
2. Only one image-to-video generation method is included in the benchmark. The robustness against other image-to-video generation methods such as [1] is not evaluated.


[1] Hu, Yaosi, Chong Luo, and Zhenzhong Chen. "Make it move: controllable image-to-video generation with text descriptions." Proceedings of the IEEE/CVF Conference on Computer Vision and Pattern Recognition. 2022.

**Questions:**

1. What is the reason for choosing only these four types of image editing methods (image regeneration, global editing, local editing, and image-to-video generation) to evaluate the image watermarking robustness, against image editing?
2. What is the motivation for using SDXL-Turbo as the generative prior for watermark encoding? If it is just to avoid multi-step sampling, there should be lots of one-step generative models to choose from, for example, the SDXS [2].

[2] Song, Yuda, Zehao Sun, and Xuanwu Yin. "SDXS: Real-Time One-Step Latent Diffusion Models with Image Conditions." arXiv preprint arXiv:2403.16627 (2024).

---

> ### Author Response · Authors · 2024-11-21
> **Response to Reviewer biAS (1/2)**
>
> We sincerely appreciate your recognition of our focus on the important and underexplored area of image watermark robustness against editing, the comprehensiveness of our benchmark, the novelty and effectiveness of our method, and the overall quality of our writing!
>
> We will address all the concerns point by point.
>
> ***
> **Weakness 1 & Question 1: What is the reason for choosing only these four types of image editing methods (image regeneration, global editing, local editing, and image-to-video generation) to evaluate the image watermarking robustness, against image editing?**
>
> Thank you for your question. We provide an explanation below and have also included it in Appendix G.1 (highlighted in red) to facilitate readers’ understanding.
>
> These four types of editing encompass the majority of editing needs. Image editing can be broadly categorized into global and local editing, as user edits typically affect either the entire image or a specific part of it.
>
> - Global editing involves altering most of an image's pixels while maintaining its overall layout, components, or semantics. Techniques such as style transfer, cartoonization, image translation, and scene transformation fall under this category and produce similar effects. For example, using prompts like "turn it into Van Gogh style" or "convert it into a sketchy painting" in an image editing model can effectively achieve style transfer.
>
> - Local editing, on the other hand, refers to modifications applied to specific elements, semantics, or regions within an image. This category includes image inpainting, image composition, object manipulation, attribute manipulation, and so forth.
>
> - While image regeneration and image-to-video generation are not strictly considered forms of image editing, they can be used to create similar digital content while removing watermarks, thereby posing a threat to copyright protection. For this reason, we have included them in our benchmark.
>
> ***
> **Weakness 2: Only one image-to-video generation method is included in the benchmark. The robustness against other image-to-video generation methods such as [1] is not evaluated.**
>
> Thank you for highlighting another image-to-video generation model (MAGE [1]). Following your recommendation, we have conducted experiments on it.
>
> Since MAGE [1] cannot process natural images, we are unable to use our benchmark images for evaluation. Instead, we utilize the CATER-GEN-v2 dataset proposed in their paper, which is a more complex version compared to CATER-GEN-v1. This dataset contains three to eight objects per video, with each object having four attributes randomly selected from five shapes, three sizes, nine colors, and two materials.
>
> We add watermarks to 1,000 images from the CATER-GEN-v2 dataset and use the MAGE+ model to perform text-image-to-video (TI2V) generation, producing 10-frame videos. For watermark detection, we analyze frames 2, 4, 6, 8, and 10. The average detection accuracies across 1,000 videos are presented in the table below and are also included in Appendix G.4 (Table 6) (highlighted in red).
>
> ***
> | Method       | Frame 2 | Frame 4 | Frame 6 | Frame 8 | Frame 10 | Average |
> |--------------|---------|---------|---------|---------|----------|---------|
> | MBRS         | 89.57   | 88.67   | 87.45   | 86.51   | 84.42    | 87.32   |
> | CIN          | 45.92   | 44.78   | 43.21   | 42.17   | 40.71    | 43.36   |
> | PIMoG        | 78.23   | 76.99   | 75.72   | 74.91   | 73.12    | 75.79   |
> | RivaGAN      | 56.87   | 54.83   | 53.21   | 52.14   | 51.01    | 53.61   |
> | SepMark      | 63.45   | 62.15   | 61.03   | 60.89   | 59.24    | 61.35   |
> | DWTDCT       | 30.57   | 29.51   | 28.89   | 27.72   | 26.87    | 28.71   |
> | DWTDCTSVD    | 38.56   | 38.54   | 37.12   | 36.81   | 35.74    | 37.35   |
> | SSL          | 81.21   | 80.95   | 78.99   | 77.18   | 76.12    | 78.89   |
> | StegaStamp   | 91.25   | 90.34   | 89.12   | 88.67   | 87.33    | 89.34   |
> | TrustMark    | 90.35   | 90.12   | 89.45   | 87.69   | 86.13    | 88.75   |
> | EditGuard    | 42.57   | 41.46   | 40.55   | 39.91   | 38.17    | 40.53   |
> | VINE-Base    | 92.22   | 91.35   | 90.74   | 89.12   | 88.01    | 90.29   |
> | VINE-Robust  | 93.14   | 92.88   | 91.32   | 90.27   | 89.12    | 91.35   |
> ***
>
> Interestingly, we found that the detection accuracies of most watermarking models are higher compared to when testing with the SVD. We attribute this to the simplicity of the dataset: the background remains mostly unchanged without significant camera motion, and only a few objects move while most remain static. This makes detection easier than in the SVD case, which typically involves camera motion effects. In this case, VINE still outperforms other watermarking models.

---

> ### Author Response · Authors · 2024-11-21
> **Response to Reviewer biAS (2/2)**
>
> **Question 2: What is the motivation for using SDXL-Turbo as the generative prior for watermark encoding? If it is just to avoid multi-step sampling, there should be lots of one-step generative models to choose from, for example, the SDXS [2].**
>
> The motivation for using one-step diffusion models is to assess whether a powerful generative prior enhances watermarking performance. We chose SDXL-Turbo as the first choice because we believe it is a highly representative one-step generative model in all, and our work currently takes the initial steps to verify whether a commonly used generative prior is beneficial for watermarking.
>
> We appreciate that you suggested using other potentially better one-step or few-step models, such as SDXS, LCM, InstaFlow, SD3.5, SwiftBrush, DMD, UFOGen, or Flux.Shenell. Although our SDXL-Turbo-based model has already advanced watermarking performance, we believe investigating alternative models with better prior performance could be a valuable direction for future research!
>
> ***
> We hope this addresses any concerns you may have. If you feel further improvements are necessary, we would be happy to make additional revisions!
>
> ***
> [1] Hu, Yaosi, Chong Luo, and Zhenzhong Chen. "Make it move: controllable image-to-video generation with text descriptions." Proceedings of the IEEE/CVF Conference on Computer Vision and Pattern Recognition. 2022.
>
> [2] Song, Yuda, Zehao Sun, and Xuanwu Yin. "SDXS: Real-Time One-Step Latent Diffusion Models with Image Conditions." arXiv preprint arXiv:2403.16627 (2024).

---

> > ### Comment · Reviewer_biAS · 2024-11-22
> >
> > I think all of my concerns are well addressed. I would like to raise my rating.

---

> > > ### Author Response · Authors · 2024-11-22
> > >
> > > Thank you for your prompt response and positive feedback on our revisions. Please let us know if you have any further questions or suggestions.
> > >
> > > We also appreciate your valuable contributions throughout the review process!

---

### Official Review · Reviewer_swi8 · 2024-11-03

**Soundness:** 2
**Presentation:** 3
**Contribution:** 3
**Rating:** 6
**Confidence:** 4

**Summary:**

This paper introduces a new evaluation benchmark, W-Bench, designed to test the robustness of image watermarking methods under image editing supported by large-scale generative models. W-Bench includes image regeneration, global editing, local editing, and image-to-video generation. The authors also propose VINE, a watermarking method utilizing generative priors to enhance the robustness and visual quality of watermark embedding. Experiments show that VINE outperforms existing watermarking methods across various image editing techniques.

**Strengths:**

1. Comprehensive Evaluation Framework: W-Bench covers a variety of image editing techniques, providing a comprehensive platform for assessing the robustness of watermarking methods.

2. Innovative Use of Generative Priors: VINE embeds watermarks by adapting pretrained large-scale generative models, making the embedding more imperceptible and robust.

3. This task is innovative, focusing on watermarking that is robust against image editing methods.

**Weaknesses:**

TreeRing, Gaussian Shading, and RingID, which add watermarks in the frequency domain of the initial noise, are generally considered robust against image editing (e.g., prompt2prompt) and regeneration. This paper lacks this crucial comparison. If these methods are also robust to image editing, the contribution of this paper may be diminished.

Reference:
1. Tree-ring watermarks: Fingerprints for diffusion images that are invisible and robust
2. Ringid: Rethinking tree-ring watermarking for enhanced multi-key identification
3. Gaussian Shading: Provable Performance-Lossless Image Watermarking for Diffusion Models

**Questions:**

1. I have doubts about the results in Figure 5(a). The experimental results show that 250-step noise in image regeneration can significantly disrupt the watermark（bit acc). Does this mean that global image editing (e.g., SDedit, prompt2prompt) with 250 steps can also completely remove the watermark? If so, I believe this result does not demonstrate robustness, as global image editing often uses even more denoising steps.

---

> ### Author Response · Authors · 2024-11-21
> **Response to Reviewer swi8 (1/2)**
>
> We sincerely appreciate your recognition of the innovation in our task and the use of generative priors, as well as your acknowledgment of the comprehensiveness of our evaluation benchmark!
>
> We will address all the concerns point by point.
>
> ***
> **Weakness 1: TreeRing, Gaussian Shading, and RingID, which add watermarks in the frequency domain of the initial noise, are generally considered robust against image editing (e.g., prompt2prompt) and regeneration. This paper lacks this crucial comparison. If these methods are also robust to image editing, the contribution of this paper may be diminished.**
>
> Thank you for your insightful questions. TreeRing, Gaussian Shading, and RingID are well-known in-generation watermarking techniques designed for watermarking generated images. However, these methods are not applicable to real images and therefore cannot be used for copyright protection of authentic photographs.
>
> Experiments are conducted to verify this. Specifically, we apply DDIM/DPM inversion techniques to extract the initial noise from a real image, add a watermark using these methods, and then invert back to obtain the image, the resulting image would differ significantly from the original one, undermining the photographer’s intent to protect their work. The visual results are shown in Figure 6 of the new version of our submission.
>
> Consequently, while these methods are highly effective for watermarking generated images, they fall outside the scope of our study. All of our baseline methods are capable of adding watermarks to real images. We have also included a discussion of these well-known in-generation watermarking methods in Appendix A.1. We kindly invite you to review that section, and we hope this addresses the concerns you may have on this matter. Your feedback is greatly appreciated!

---

> ### Author Response · Authors · 2024-11-21
> **Response to Reviewer swi8 (2/2)**
>
> **Question 1: I have doubts about the results in Figure 5(a). The experimental results show that 250-step noise in image regeneration can significantly disrupt the watermark（bit acc). Does this mean that global image editing (e.g., SDedit, prompt2prompt) with 250 steps can also completely remove the watermark?**
>
> We apologize for any confusion.
>
> The key point is that an imperfect bit accuracy of approximately 80% does not indicate that the watermark has been compromised. Even if an image achieves an extracted bit accuracy of around only 80%, it still has a very high probability—potentially exceeding 99%—of being flagged as watermarked in the statistical test. Therefore, the 250-step noise applied during image regeneration does not disrupt the watermark.
>
> The following provides a detailed introduction to the statistical test.
> ***
> Let $\boldsymbol{w} \in \\{0,1\\}^k$ be the $k$-bit ground-truth watermark. The watermark $\boldsymbol{w}^\prime$ extracted from a watermarked image $\boldsymbol{x}_w$ is compared with the ground-truth $\boldsymbol{w}$ for detection. The detection statistical test relies on the number of matching bits $M(\boldsymbol{w}, \boldsymbol{w}^\prime)$: If
> $$
> M(\boldsymbol{w}, \boldsymbol{w}^\prime) \ge \tau, \text{where} \ \tau \in \\{0,1,2, \cdots, k\\},
> $$
> then the image is flagged as watermarked. Formally, we test the statistical hypothesis $H_1$: '$\boldsymbol{x}$ contains the watermark $\boldsymbol{w}$' against the null hypothesis $H_0$: '$\boldsymbol{x}$ does not contain the watermark $\boldsymbol{w}$'. Under $H_0$ (i.e., for original images), if the extracted bits $\boldsymbol{w} = \\{w_1^\prime, w_2^\prime, \cdots, w_k^\prime\\}$ (where $w_i^\prime$ is the i-th extracted bit) from a model are independent and identically distributed (i.i.d.) Bernoulli random variables with the matching probability $p_o$, then $M(\boldsymbol{w}, \boldsymbol{w}^\prime)$ follows a binomial distribution with parameters $(k, p_o)$. This assumption is verified by [1].
>
> The false positive rate (FPR) is the probability that $M(\boldsymbol{w}, \boldsymbol{w}^\prime)$ takes a value bigger than the threshold $\tau$ under the null hypothesis $H_0$. It is obtained from the CDF of the binomial distribution, and a closed-form can be written with the regularized incomplete beta function $I_p(a,b)$ [1]:
> $$
> \text{FPR}(\tau) = \mathbb{P} \left( M(\boldsymbol{w}, \boldsymbol{w}^\prime)>\tau|H_0 \right) = \sum_{i=\tau + 1}^{k} \binom{k}{i} p_o^i(1-p_o)^{k-i} = I_{p_o}(\tau + 1, k - \tau),
> $$
> where under $H_0$ (i.e., images without watermark $\boldsymbol{w}$), $p_o$ should ideally be close to 0.5 to minimize the risk of false positive detection.
>
> Similarly, the true positive rate (TPR) represents the probability that the number of matching bits exceeds the threshold $\tau$ under $H_1$, where the image contains the watermark. Thus, the TPR can be calculated by:
> $$
> \text{TPR}(\tau) = \mathbb{P} \left( M(\boldsymbol{w}, \boldsymbol{w}^\prime)>\tau | H_1 \right) = \sum_{i=\tau + 1}^{k} \binom{k}{i} p_w^i(1-p_w)^{k-i} = I_{p_w}(\tau + 1, k - \tau),
> $$
> where under $H_1$ (i.e., images with watermark $\boldsymbol{w}$), $p_w$ should ideally be high enough (e.g., exceeding 0.8) to ensure the effectiveness of a watermarking model.
>
> To further demonstrate that _neither high bit accuracy nor AUROC alone guarantees a high TPR at a low FPR_, consider the following example. Suppose we have a 100-bit watermarking model with a threshold $\tau$ of 70 to determine whether an image contains watermark $\boldsymbol{w}$. If the model extracts bits from watermarked images with a matching probability $p_w = 0.8 $ and from original images with a matching probability $p_o = 0.5 $, the resulting FPR would be $ 1.6 \times 10^{-5} $ and the TPR would be 0.99. In this scenario, even though the bit accuracy for watermarked images is not exceptionally high (e.g., below 0.9), the model still achieves a high TPR at a very low FPR. In contrast, if another model has $ p_w = 0.9 $ and $ p_o = 0.7 $, achieving the same FPR would require setting the threshold $\tau$ to 87. Under these conditions, the TPR would only be 0.8. This example demonstrates that high bit accuracy for watermarked images does not necessarily ensure a high TPR when maintaining a low FPR. Therefore, relying solely on bit accuracy or AUROC may not be sufficient for achieving the desired performance in watermark detection.
>
> ***
> We have included this information in Appendix C in the new version of our submission (highlighted in blue) to enhance reader understanding. We kindly invite you to review it and hope that it addresses the concerns you may have regarding this matter. If you believe further improvements are needed, we would be happy to make additional revisions. Thank you for helping to make our work more complete!
>
> [1] The stable signature: Rooting watermarks in latent diffusion models.

---

> > ### Author Response · Authors · 2024-11-26
> >
> > Dear Reviewer swi8,
> >
> > We sincerely appreciate the time and effort you have dedicated to reviewing our paper.
> >
> > As the discussion period approaches its end, we would like to gently remind you that there are seven days remaining for any additional comments or questions. We would be grateful for the opportunity to address any further concerns you may have before the discussion phase concludes.
> >
> > Thank you very much!
> >
> > Many thanks,
> >
> > The Authors

---

> > ### Comment · Reviewer_swi8 · 2024-11-26
> >
> > Thank you for the detailed response, which has essentially resolved my concern. I am willing to increase the score to 7.

---

> ### Author Response · Authors · 2024-11-26
>
> Dear Reviewer swi8,
>
> Thank you very much for your positive feedback on our revisions! We are delighted to hear that your concerns have been essentially addressed.
>
> As the current scoring system does not include an option for a score of 7, we kindly ask if you would consider adjusting your score to 8 in the system, if possible.
>
> Should you have any further questions or suggestions, please do not hesitate to let us know. We greatly appreciate your valuable contributions throughout the review process!
>
> Warm regards,
>
> The Authors

---

> > ### Comment · Reviewer_swi8 · 2024-12-01
> >
> > OK，I am willing to increase the score to 8.

---

> > > ### Author Response · Authors · 2024-12-01
> > >
> > > Dear Reviewer swi8,
> > >
> > > We would like to express our gratitude for your valuable feedback on our submission and for indicating your willingness to increase the score to 8. Your support is greatly appreciated!
> > >
> > > We noticed that the score adjustment hasn't been reflected in OpenReview yet. If it's not too much trouble, could you kindly update the score by editing your review in the OpenReview system?
> > >
> > > Thank you once again for your time and valuable feedback!
> > >
> > > Warm regards,
> > >
> > > The Authors

---

### Official Review · Reviewer_QuDW · 2024-11-03

**Soundness:** 3
**Presentation:** 3
**Contribution:** 3
**Rating:** 6
**Confidence:** 4

**Summary:**

This paper introduces W-Bench, the first comprehensive benchmark designed to evaluate the robustness of watermarking methods against a wide range of image editing techniques, including image regeneration, global editing, local editing, and image-to-video generation. Authors reveal that image editing and blurring distortion predominantly remove watermarking patterns in high-frequency bands, while those in low-frequency bands remain less affected. Based on this, distortions are used as surrogate attacks to overcome the challenges of using T2I models during training and to enhance the robustness of the watermark. The authors approach the watermark encoder as a conditional generative model and introduce two techniques to adapt SDXL-Turbo, a pretrained one-step T2I model, for the watermarking task. Experimental results demonstrate that VINE is robust against multiple image editing methods while maintaining high image quality.

**Strengths:**

1.	The proposed method is easy yet effective. The combination of different losses is reasonable.
2.	The validation of watermarking patterns in high-frequency bands after image editing and blurring is solid.
3.	The experimental results show the proposed watermarking method is robust enough against multiple image editing methods.

**Weaknesses:**

1.	This paper lacks the validation of hypotheses in Line 249.
2.	The task of watermarking against Image Editing seems worthless.
3.	The watermarking pattern existing in high-frequency bands after image blurring is not a new discovery. However, the author spends too much text on it.

**Questions:**

1. Although the watermarking against Image Editing is interesting and novel, I cannot get the value of this task. Can you elaborate the perspective of this task?
2. The author hypothesizes that a powerful generative prior can facilitate embedding information more invisibly while enhancing robustness (Line 249). Why hypothesize that? What are the assumptions based on?
3. What is the purpose of finetuning VINE-B to VINE-R using Instruct-Pix2Pix? (Line 323)
4. Why is the resolution not unified? (Line 1042)
5. Is VINE only work on the Image Editing task? What about other common watermarking tasks?

---

> ### Author Response · Authors · 2024-11-21
> **Response to Reviewer QuDW (1/2)**
>
> We sincerely appreciate your recognition of our method as both reasonable and effective, as well as the solidity of our validation!
>
> We will address all the concerns point by point.
> ***
> **Weakness 2 & Question 1: Although the watermarking against Image Editing is interesting and novel, I cannot get the value of this task. Can you elaborate the perspective of this task?**
>
> _The Importance of Watermarking Images Against AI-Powered Image Editing_
>
> Invisible watermarks enable creators to assert ownership of their digital content. If a watermark can be easily removed through minor alterations (such as local editing) or visually imperceptible changes (like image regeneration), malicious users could remove or add a small element (such as a dog) to an artwork and publish it without proper attribution, which could lead to copyright issues. Such non-robust watermarks pose challenges for ownership assertion.
>
> In contrast, editing-resistant watermarks can enhance security by embedding links that trace the source of leaks or unauthorized distributions back to the original distributor. For example, photographers and media companies can monitor the usage of their images, helping to identify unauthorized editing or redistribution.
> ***
> **Weakness 1 & Question 2: (1) This paper lacks the validation of hypotheses in Line 249; (2) The author hypothesizes that a powerful generative prior can facilitate embedding information more invisibly while enhancing robustness (Line 249). Why hypothesize that? What are the assumptions based on?**
>
> 1) _The validation of the hypothesis._ The validation is provided in the ablation study (Table 2 Config H). In Table 2, Config H is trained with randomly initialized weights instead of pretrained ones while retaining all other settings from Config G. Comparing Config H with Config G, it can be seen that the pretrained prior knowledge is beneficial for image quality, and thus it helps embedding information more invisibly.
>
> 2) _Rationale for the Hypothesis._ This hypothesis is based on the expectation that the output of the watermarking process should visually resemble the input image, which is a natural, artifact-free image. Powerful generative priors, particularly large-scale diffusion models, have the capability to generate images within this natural, artifact-free manifold. SDXL-Turbo, distilled from its teacher model SDXL, inherits its generative capability. Thus, we expect that initializing with this strong generative prior will benefit the training process and significantly weaken the artifacts in watermarked images.
>
> We hope this addresses the concerns you may have on this matter and remain available for further discussion!
> ***
> **Weakness 3: The watermarking pattern existing in high-frequency bands after image blurring is not a new discovery. However, the author spends too much text on it.**
>
> Thank you for your valuable suggestions. In response to your feedback, we have removed lines 212-213 of the initial version of our submission to reduce the discussion on blurring. Following your recommendation, now the emphasis has been shifted to the similarities between the frequency characteristics of image editing and blurring. This is a new discovery that we believe would be important and insightful for readers.
> ***
> **Question 3: What is the purpose of finetuning VINE-B to VINE-R using Instruct-Pix2Pix? (Line 323)**
>
> The purpose of incorporating Instruct-Pix2Pix, a representative editing model, into the finetuning process is to further boost the robustness against image editing. This integration is performed during finetuning because attempting to include the editing model at the initial training stage using a straight-through estimator (STE) results in convergence failures.
>
> In fact, even without this finetuning step, VINE-B already outperforms the baseline models overall (see Figure 1). We are interested in determining whether a well-trained watermarking model can be finetuned with an editing model through STE to further improve robustness. Therefore, we implement this variant.
> ***
> **Question 4: Why is the resolution not unified? (Line 1042)**
>
> Thanks for your question. All baseline models were evaluated using their official checkpoints, which were trained at various resolutions. Table 3 presents the performance of these methods at their original training resolutions (thus not unified). The purpose of Table 3 is to demonstrate that scaling the resolution does not affect the quality of the encoded images. The results after scaling to the unified resolution are shown in Table 1.
>
> Additionally, Figures 9 and 10 of the new version of our submission (or Figures 7 and 8 of the 1st version of our submission) illustrate that resolution scaling does not impact detection robustness. These evaluations were conducted to determine whether resolution scaling affects the original performance, as it is necessary to scale images to a unified 512 × 512 resolution before inputting them into the editing models.

---

> ### Author Response · Authors · 2024-11-21
> **Response to Reviewer QuDW (2/2)**
>
> **Question 5: Is VINE only work on the Image Editing task? What about other common watermarking tasks?**
>
> Thank you very much for highlighting this important question! We fully agree with the significance of this aspect of our method. VINE also remains robust against common degradations such as JPEG compression, Gaussian noise, contrast modifications, brightness adjustments, and more. For a comprehensive overview of its performance, we kindly direct your attention to Figure 9 in the revised version of our submission (previously Figure 7). We sincerely hope this helps address your concerns, and we are always available to discuss this further if needed. Your feedback is greatly appreciated!
> ***
>
> If the reviewer believes further improvements are needed, we would be happy to make additional revisions!

---

> > ### Author Response · Authors · 2024-11-26
> >
> > Dear Reviewer QuDW,
> >
> > We sincerely appreciate the time and effort you have dedicated to reviewing our paper.
> >
> > As the discussion period approaches its end, we would like to gently remind you that there are seven days remaining for any additional comments or questions. We would be grateful for the opportunity to address any further concerns you may have before the discussion phase concludes.
> >
> > Thank you very much!
> >
> > Many thanks,
> >
> > The Authors

---

> ### Comment · Reviewer_QuDW · 2024-11-26
>
> Thank you for your response. I would like to keep the previous score.

---

> > ### Author Response · Authors · 2024-11-26
> >
> > Dear Reviewer QuDW,
> >
> > Thank you for dedicating your time and effort to reviewing our paper!
> >
> > If you have any further concerns, please do not hesitate to let us know. We would greatly appreciate the opportunity to address any additional issues you may have before the discussion phase concludes.
> >
> > Warm regards,
> >
> > The Authors

---

### Official Review · Reviewer_3J7W · 2024-11-04

**Soundness:** 2
**Presentation:** 3
**Contribution:** 2
**Rating:** 6
**Confidence:** 2

**Summary:**

This paper presents VINE, a watermarking method designed to withstand various image editing techniques enabled by advanced generative models. It also introduces W-Bench, a benchmark that evaluates watermark robustness against multiple types of edits, making it a valuable resource for watermarking research.

**Strengths:**

- The paper is clearly written and organized, with effective figures explaining both W-Bench and VINE.

- The paper provides rigorous evaluations, testing VINE and eleven other watermarking models on diverse editing techniques.

**Weaknesses:**

- EditGuard is primarily designed for editing detection, not robust watermarking, and it was not tested with its most robust configuration. This impacts the fairness of the evaluation, as EditGuard’s focus and strengths differ from VINE’s intended use.

**Questions:**

See weakness.

---

> ### Author Response · Authors · 2024-11-21
> **Response to Reviewer 3J7W**
>
> We sincerely appreciate your recognition of our paper’s well organization and rigorous evaluation!
>
> ***
> **Weaknesses 1: EditGuard is primarily designed for editing detection, not robust watermarking, and it was not tested with its most robust configuration. This impacts the fairness of the evaluation, as EditGuard’s focus and strengths differ from VINE’s intended use.**
>
> We appreciate your feedback and have updated Section 4.3 (lines 470-472) in the new submission accordingly. A new remark, highlighted in purple, has been added to clarify the primary focus of EditGuard's design. For your convenience, the remark is also provided below.
>
> EditGuard is not designed for robust watermarking against image editing, as it is trained with mild degradation. Instead, it offers a feature for tamper localization, enabling the identification of edited regions.
>
> ***
> If you believe further improvements are needed, we would be happy to make additional revisions!

---

> > ### Comment · Reviewer_3J7W · 2024-11-21
> > **Practical value**
> >
> > Thanks for the feedback. After reading the review of QuDW, I also started to doubt about the practical value of this field.
> >
> > Could the authors provide an example of a network-based method being practically applied? My research area is not in this field, but it seems to me that this field offers little practical value beyond papers and some handcraft metrics.

---

> ### Author Response · Authors · 2024-11-21
> **Response about practical value**
>
> Thank you for your question. Here are some practical examples of robust invisible watermarking in use:
>
> - IMATAG: They aim to protect businesses from the unauthorized use of their images and videos. Visit their [homepage](https://www.imatag.com/) or check out their [X profile](https://x.com/imatag) for more information. In short, by embedding their robust invisible watermarks and utilizing their detector, IMATAG can (1) identify who is leaking your visual content; (2) monitor who is using your visual content; (3) verify the authenticity of your visual content.
>
> - Adobe Integration: IMATAG is also integrated with Adobe software [here](https://exchange.adobe.com/apps/cc/101789/imatag-invisible-watermark-and-image-monitoring).
>
> - Adobe Content Authenticity: It is designed to help creators protect and receive attribution for their work with Content Credentials. Content Credentials combine digital fingerprinting, invisible watermarking and cryptographically signed metadata, helping to ensure that Content Credentials remain intact and verifiable across the digital ecosystem [[blog link]](https://news.adobe.com/news/2024/10/aca-announcement).
>
> - Google SynthID: Google uses its [SynthID](https://deepmind.google/discover/blog/identifying-ai-generated-images-with-synthid/) for Vertex AI customers.
>
> Please note that these technologies are not open-sourced. While they can protect digital content against common transformations, their robustness against AI-powered image editing is underexplored.
>
> If a malicious user uses generative models to remove or add a small element (such as a dog) to an artwork and publishes it without proper attribution, it could lead to copyright issues.

---

> > ### Author Response · Authors · 2024-11-26
> >
> > Dear Reviewer 3J7W,
> >
> > We sincerely appreciate the time and effort you have dedicated to reviewing our paper.
> >
> > As the discussion period approaches its end, we would like to gently remind you that there are seven days remaining for any additional comments or questions. We would be grateful for the opportunity to address any further concerns you may have before the discussion phase concludes.
> >
> > Thank you very much!
> >
> > Many thanks,
> >
> > The Authors

---

### Comment · Area_Chair_vAuG · 2024-11-25

Hi Reviewers,

We are approaching the deadline for author-reviewer discussion phase. Authors has already provided their rebuttal. In case you haven't checked them, please look at them ASAP. Thanks a million for your help!

---

### Meta-Review · Area_Chair_vAuG · 2024-12-20

**Metareview:**

This paper is working on robust watermarking. Authors first proposed W-Bench to evaluate the robustness of watermarking methods against a wide range of image editing techniques. Then proposed VINE to improve watermarking robustness against all these different edits. Experimental results show effectiveness of the proposed methods.

5 reviewers unanimously considered this paper above the acceptance bar.

Strengths of this paper are: 1) clearly written and organized; 2) rigorous and solid evaluations; 3) proposed method is easy yet effective; 4) task is innovative and important.

Weaknesses are: 1) unfair comparison for EditGuard; 2) lacks validation of some hypotheses; 3) lacks comparison; 4) need more experiments, etc;

After rebuttal, reviewers' concerns are addressed. Some reviewers increased scores and some kept the scores. Given these, AC decide to accept this paper.

**Additional Comments On Reviewer Discussion:**

After rebuttal, reviewers' concerns are addressed. Some reviewers increased scores and some kept the scores. Given these, AC decide to accept this paper.

---

### Decision · Program_Chairs · 2025-01-22

Accept (Poster)